# Continuous Treatment with IncobotulinumtoxinA Despite Presence of BoNT/A Neutralizing Antibodies: Immunological Hypothesis and a Case Report

**DOI:** 10.3390/toxins16100422

**Published:** 2024-10-01

**Authors:** Michael Uwe Martin, Clifton Ming Tay, Tuck Wah Siew

**Affiliations:** 1Independent Researcher, 31832 Springe, Germany; 2Merz Asia Pacific Pte., Ltd., Singapore 138567, Singapore; clifton.tay@merz.sg; 3Radium Medical Aesthetics, 3 Temasek Boulevard #03-325/326/327/328, Suntec City Mall, Singapore 038983, Singapore

**Keywords:** immunologic adjuvants, Botulinum Neurotoxin A, complexing protein free, immunoresistance, immunological memory, IncobotulinumtoxinA, reactivation, secondary non-response, therapy failure

## Abstract

Botulinum Neurotoxin A (BoNT/A) is a bacterial protein that has proven to be a valuable pharmaceutical in therapeutic indications and aesthetic medicine. One major concern is the formation of neutralizing antibodies (nAbs) to the core BoNT/A protein. These can interfere with the therapy, resulting in partial or complete antibody (Ab)-mediated secondary non-response (SNR) or immunoresistance. If titers of nAbs reach a level high enough that all injected BoNT/A molecules are neutralized, immunoresistance occurs. Studies have shown that continuation of treatment of neurology patients who had developed Ab-mediated partial SNR against complexing protein-containing (CPC-) BoNT/A was in some cases successful if patients were switched to complexing protein-free (CPF-) incobotulinumtoxinA (INCO). This seems to contradict the layperson’s basic immunological understanding that repeated injection with the same antigen BoNT/A should lead to an increase in antigen-specific antibody titers. As such, we strive to explain how immunological memory works in general, and based on this, we propose a working hypothesis for this paradoxical phenomenon observed in some, but not all, neurology patients with immunoresistance. A critical factor is the presence of potentially immune-stimulatory components in CPC-BoNT/A products that can act as immunologic adjuvants and activate not only naïve, but also memory B lymphocyte responses. Furthermore, we propose that continuous injection of a BoN/TA formulation with low immunogenicity, e.g., INCO, may be a viable option for aesthetic patients with existing nAbs. These concepts are supported by a real-world case example of a patient with immunoresistance whose nAb levels declined with corresponding resumption of clinical response despite regular INCO injections.

## 1. Introduction and Background

### 1.1. Immunoresistance Is a Major Concern and Limitation of BoNT/A Therapy

BoNT/A is a bacterial protein that has proven to be a most valuable pharmaceutical in therapeutic indications and aesthetic medicine for more than 30 years now. However, right from the beginning of BoNT/A therapies, one major concern was the possible induction of antibodies (Abs) to the bacterial protein [1]. The clinical effects of BoNT/A typically last for months and BoNT/A has to be re-injected at regular time intervals, frequently for many years, if treatment results are to be maintained. This schedule is strongly reminiscent of a vaccination protocol. Abs to BoNT/A products may be non-neutralizing or neutralizing, but only the latter are clinically relevant as they block and inhibit the neurotoxic effect mediated by the 150 kDa core BoNT/A molecule, resulting in Ab-mediated non-response or immunoresistance. In this article, we use the term “immunoresistance” to describe non-responsiveness to BoNT/A treatment due to the presence of nAbs to the core 150 kDa BoNT/A molecule. Additionally, one must clearly differentiate primary and secondary immunoresistance. Primary immunoresistance is rarely seen in patients (reviewed in [2]), and the existence of nAbs is commonly ascribed to an unknowing exposure of such persons to BoNT/A in the past, e.g., by food poisoning, or active immunization (vaccination). These persons do not have any clinical response to BoNT/A injections right from the start. Secondary immunoresistance, in contrast, describes the situation in which a patient had initially responded well to the treatment, but the desired effects started to diminish over time. This phenomenon is well documented in neurological indications (reviewed in [3,4,5,6,7]) and in cosmetic medicine (reviewed in [8,9,10,11]). While nAbs to BoNT/A are the cause of secondary treatment failure in approximately 50% of cases (reviewed in [3]), other reasons for therapy failure (primary and secondary) have been discussed, including disease progression, insufficient dosage, inadequate handling of the product (inadequate reconstitution, storage), targeting incorrect muscles, improper injection technique, but also discordance in perception of benefit between patient and health care professional (HCPs).

If immunoresistance occurs, HCPs have to either increase the dose of BoNT/A and/or shorten the timing between injections to try to achieve the initial outcome. However, titers of nAbs have increased in some patients to such a level that all injected BoNT/A molecules are neutralized before they can be taken up into the nerve cells. Complete therapy failure is the consequence (reviewed in [2]). In such a situation, the treating physician does not have many remaining options (reviewed in [12]). Switching patients to BoNT/B is a possibility but tends to be only a short-term alternative due its more pronounced immunogenicity compared to BoNT/A, with a high proportion of patients eventually developing nAbs to BoNT/B [13] and reviewed in [3,4]. Another option for patients with complete immunoresistance is to suspend BoNT/A injections for several years [14], euphemistically named a “treatment holiday”. While this may seem acceptable in elective aesthetic applications, it always means a dramatic loss of quality of life for patients with neurological disorders.

### 1.2. Differences in Immunogenicity of BoNT/A Products Correlate with the Presence of Complexing Proteins and Other Bacterial Components

Differences exist in the immunogenicity of BoNT/A products, three of which have been investigated in detail in the past decade, namely onabotulinumtoxinA (ONA), abobotulinumtoxinA (ABO), and incobotulinumtoxinA (INCO). All three products contain the core 150 kDa neurotoxin BoNT/A. This consists of one heavy and one light chain covalently linked by a disulfide bridge and is the only component necessary and sufficient to confer neurotoxicity (reviewed in [15,16]). The products differ in proprietary manufacturing protocols and the presence of complexing proteins, other bacterial components, and excipients [17,18]. ONA and ABO are complexing protein-containing formulations [19,20]. INCO is the first commercially available complexing protein-free formulation [21,22]. It contains the highly purified and bioactive BoNT/A molecule and no other bacterial components [23,24]. Several reports demonstrate that the presence of complexing proteins and other bacterial components such as flagellin (contained in ABO [19,25]) or clostridial DNA (contained in ONA [26]) correlate with a higher risk of inducing Abs to BoNT/A (summarized in [27]) as these components may act as immune stimulators or immunologic adjuvants (see below). Several clinical studies demonstrate that immunoresistance did not occur when patients were treated exclusively with CPF-INCO even at very high doses [28,29,30,31,32,33].

### 1.3. Breaking Immunoresistance Is Possible with a Low Immunogenic CPF-BoNT/A Product

Long-term studies have shown that it requires many years of suspending BoNT/A injections before titers of nAbs in complete non-responders drop to a level where they will not interfere with a new BoNT/A injection anymore [14]. Restarting therapy after a treatment holiday with the immunogenic CPC-BoNT/A product that initially caused the generation of nAbs carries a high risk of reactivating the immunological memory to again produce nAbs [14,34]. On the contrary, restarting the therapy with low-immunogenic CPF-INCO was successful and did not stimulate further production of nAbs [33,35]. It was concluded that switching from CPC-BoNT/A formulations to CPF-INCO represents a treatment option following the resolution of nAb titers after many years of treatment cessation [33,34]. This led to the next question on what would happen if such patients were immediately switched from a CPC-BoNT/A product to CPF-INCO without a pause. The expectation was that BoNT/A, irrespective of the product injected, would not work as long as nAb titers were high enough to neutralize all BoNT/A core neurotoxin molecules present. But would there be any difference in the trend of nAb titers with continuous BoNT/A injections due to differences in immunogenicity of the individual products?

Hefter, et al. addressed this issue in a single-cohort 4-year follow-up study of 37 cervical dystonia patients (=100%) who had developed partial immunoresistance against the CPC-BoNT/A formulations ONA and / or ABO [36]. Without interrupting treatment, some of the partial non-responders were switched to receive CPF-INCO every 3 months and nAb titers were monitored closely for the following years. This cohort was compared to another one in which BoNT/A treatment was discontinued (n = 24). Blood was drawn at the beginning of the study and then yearly to test for the presence of nAbs to BoNT/A using the mouse hemidiaphragm assay (MHDA) [37,38,39]. First, it must be emphasized that strong inter-individual differences were observed in nAb titers over time, as was to be expected in unrelated human beings. There was a gradual decline in nAb titers after discontinuing BoNT/A injections in most of the patients in the “control” group over the study period of 4 years, confirming results from earlier studies [14]. Basically, three types of nAb titer responses were observed in the 37 in patients with partial immunoresistance who were switched to CPF-INCO: unchanged titers, continuously declining titers, and transient increase in nAb titers with subsequent decrease. In six patients (16%) nAb titers more or less remained unchanged, indicating that the adaptive immune response in these patients had already been maximally stimulated by prior repeated injections of CPC-BoNT/A products. This suggests that BoNT/A-specific immunological memory lasted longer than the study period of 4 years in these patients. In 31 patients (84%), nAb titers dropped below the initial titers at the end of the observation period. In 23 patients (62%), nAb titers even reached the lower detection limit of the MHD assay or became negative.

However, nAb titers in 10 of these patients (27%) transiently increased in the first 2 years of INCO treatment before they started to decline [36]. The transient increase in nAb titers is not surprising considering that CPC- and CPF-products all contain an identical antigen—BoNT/A. This shows that, in principle, INCO can booster an existing immune response that had been induced by CPC-BoNT/A products, an observation that was confirmed by another case [40]. The time frame of cessation of nAb production was comparable to that observed in the control group in which BoNT/A injections were discontinued at the beginning of the study. Although correlating nAb titers with clinical outcome was reported to be difficult [41], improvement was detected in a subsequent study of patients with cervical dystonia who had developed partial immunoresistance to CPC-BoNT/A formulations. These patients were also switched to CPF-INCO and monitored for a period of 48 weeks [41]. An increase was observed in 23% of patients, while a constant or decreasing level was ascertained in 77% of the patients despite continuous injections with CPF-INCO. In this study, a significant improvement in cervical dystonia symptoms was observed at 48 weeks after switching to CPF-INCO. The authors concluded that despite partial secondary non-response (SNR) due to nAb formation, uninterrupted continuation of BoNT/A treatment with a low immunogenic BoNT/A product, e.g., CPF-INCO, offers a possible treatment option. A similar case was reported for a patient with musician’s dystonia [42].

### 1.4. Aims of Article

A common notion is that repeated injection with the same antigen, also called “boosting or boostering”, should rather lead to an increase and not to a decrease of antigen-specific Ab titers, a process exploited to improve protection by repeated vaccinations with the antigen. Thus, these findings seem to contradict the layperson’s understanding of how the immune system, or more precisely, the immunological memory, works. As this treatment option is of great clinical relevance for patients suffering from non-responsiveness to BoNT/A, it is pivotal to contextualize these clinical observations to our knowledge of the human immune system and try to understand the underlying immunological mechanisms.

Our first aim is to provide a general overview on how humoral immunological memory is induced and maintained, so as to prepare a simplified basis and scientific reasoning for this contradictory phenomenon. We describe how immunologic adjuvants are required in two essential steps on the way to antibody production in the naïve situation; and we call to attention the fact that memory B lymphocytes also carry receptors for immunologic adjuvants and are more easily/readily activated in a recall situation by a combination of antigen plus immunologic adjuvants than by antigen alone.

Our second objective is to explain how individual steps of activation of the immune system may apply to BoNT/A treatment. To that end, we will discuss what is known about these individual steps with respect to BoNT/A. We propose that the presence of complexing proteins and other bacterial components contained in some BoNT/A products may serve as immunologic adjuvants. This provides an explanation why memory B lymphocytes specific for BoNT/A are not or much less likely to be reactivated by a low-immunogenic CPF-BoNT/A formulation that lacks components with immunologic adjuvant properties compared to CPC-BoNT/A products.

Finally, we introduce an aesthetic patient who had developed nAb-mediated SNR during treatment with CPC-BoNT/A products as a real-world case example to illustrate these immunological concepts. This patient continued to receive CPF-INCO at regular intervals, and the nAbs dropped to reach non-detectable levels after 4 years when the patient started to respond to BoNT/A again.

## 2. How the Immune System Is Activated—The Naïve Situation

The human immune system consists of innate and adaptive arms. Both parts cooperate in a strictly hierarchical order to allow Ab production to a novel challenge, such as microbes (reviewed in [43]). Adherence to this fixed course of activation steps guarantees that a laborious Ab response is initiated only when necessary and adequate.

### 2.1. How the Immune System Becomes Activated to Produce Antibodies

In the naïve situation, a person is exposed to a challenge for the first time. Complex challenges such as microbes consist of many potential antigens. By definition, antigens are substances, frequently proteins, that activate the immune system to produce antibodies that bind specifically to this substance. Normally, dendritic cells (DCs), members of the first line of defense of the innate immune system, are the only leukocytes that act as professional APCs upon initial exposure to a (microbial) challenge (reviewed in [44,45,46]). DCs are sentinel cells of the innate immune system distributed all over the body with especially high numbers in barrier tissues such as skin or mucosa (reviewed in [47]). They are perfectly equipped with a plethora of receptors capable of recognizing molecular patterns or signatures typical of microbes that are uncommon or non-existing in human beings. Pattern-recognition receptors (PRR) on DCs include the prominent family of the Toll-like receptors (TLRs), lectins, complement receptors, scavenger receptors, and many more (reviewed in [48]). Antigen presentation via major histocompatibility complex class II (MHC II—also called human leukocyte antigen (HLA) in human beings)—is the typical type of response to an extracellular challenge such as bacteria or a bacterial protein. For this article, we will restrict the discussion here to antigen presentation via MHC II in context to BoNT/A.

### 2.2. Optimal Activation of Dendritic Cells Is Mandatory for Professional Antigen Presentation

Pattern-recognition receptors (PRRs) are activated upon encountering microbes or microbial structures (reviewed in [49]. Prototypic ligands of PRR can be considered as danger signals (reviewed in [50,51,52]) that activate innate immunity, especially DCs. These activating signals act as immunologic adjuvants, or “immune response enhancers”, substances well-known from vaccination that help to generate a strong(er) immune response to a weak antigen or vaccine (“immunologic adjuvant” defined in [53]). In a normal infection, several different types of pattern-recognition receptors will be engaged in parallel, frequently by crosslinking on the surface. In consequence, a specific pathogen will activate several different and discrete signal-transduction pathways in local DCs (reviewed in [54]) that can be considered as signal collectors or signal integrators (reviewed in [55,56]). If the integrated sum of these signals is strong enough, DCs become fully activated.

Immediately, several measures are initiated by activated DCs to face the recognized (microbial) challenge. These include the release of anti-microbial substances to harm extracellular microbes or even kill them locally before they can multiply and spread. In addition, alarm mediators (such as arachidonic acid metabolites, chemokines, and cytokines) are produced and released into the vicinity. Together with activated resident macrophages, DCs start to organize a local acute inflammatory response, including the recruitment of leukocytes from the bloodstream into the tissue. In parallel, wound-healing processes are initiated by cytokines. It is important to note that this stereotypic reaction of initiating a local inflammatory response always takes place after alarming of sentinel cells. However, it can be locally confined to this site and limited in time. It does not necessarily have to result in a clinical manifestation of inflammatory signs. If, however, DCs are fully activated, they will engulf the registered microbes or their remains and phagocytose them. Phagocytosis delivers the pathogens into the specialized compartments or phagosomes, where they are killed by anti-microbial substances and digested by enzymes. One result is the cleavage of phagocytosed microbial proteins (antigens) into peptides. Within the phagosomes, peptides are loaded onto MHC II, which serve as specialized “peptide presenting molecules”. Subsequently, the complexes of MHC II and bound antigenic peptides are transported to the plasma membrane, where they are exposed to the exterior of the DCs (Figure 1A). By this mechanism, a representative selection of antigen-specific samples is made “visible” to the T cell antigen receptor of T helper (Th) lymphocytes. In summary, fully activated DCs develop into professional APCs that link the innate and adaptive immune system by informing naïve T lymphocytes on the location, type, and severity of the microbial challenge. Activated DCs leave their sentinel position in the periphery and move to the next draining lymph node (reviewed in [57]), where they meet T lymphocytes.

### 2.3. Professional Antigen Presentation Is Mandatory for Activation and Clonal Expansion of Antigen-Specific Naïve T Helper Lymphocytes

Fully activated DCs leave the tissue and migrate to the first draining lymph node via the afferent lymphatics and function as professional APCs (rolling blue arrow in Figure 1). Concurrently, naïve T helper (Th) lymphocytes enter the lymph node from the circulation. Here, they encounter the presented peptides in MHC II on the surface of APCs. Due to their full activation, APCs also express several costimulatory molecules on their surface and, finally, release cytokines. Upon recognizing the presented peptide in MHC II with its T cell antigen receptor, this naïve antigen-specific Th cell becomes activated (Figure 1B). However, full activation is only possible if costimulatory molecules and cytokines from APCs are also present. Under these tightly controlled conditions, this activated antigen-specific Th cell starts to proliferate. In a few days, thousands of clonal offspring are generated in this lymph node. All of them retain the antigen specificity of the original antigen-specific naïve Th cell (Figure 1B). These effector cells contribute to the ongoing immune reaction. Some of them develop into memory T cells that are normally dormant and in a state of rest but can be reactivated upon re-encountering their antigen in a recall situation (reviewed in [58,59]).

### 2.4. Activation of Antigen-Specific B Lymphocytes Is Controlled by Antigen-Specific Effector T Helper Lymphocytes and Enhanced by Immunologic Adjuvants

Hundreds of thousands of naïve B cells circulate through the blood and enter lymph nodes constantly exactly where APCs support activation and clonal expansion of antigen-specific Th cells. Each B cell possesses many copies of a B cell antigen receptor (BCR), which is a plasma membrane-anchored antibody with one individual discrete specificity per B cell. Upon specific binding of an antigen, BCRs are crosslinked on the surface of this antigen-specific B cell (Figure 1C). Crosslinking leads to an activation that sparks internalization of the antigen bound to the BCR. It is important to understand that activation of a naïve B cell is optimal when other receptors are also engaged in addition to crosslinking of many copies of BCRs by antigen (reviewed in [60]). These BCR co-receptors also include pattern-recognition receptors such as TLRs (reviewed in [61,62,63]) that bind immunologic adjuvants. Parallel activation is a safety feature that ensures that optimal activation is achieved only in the presence of microbial dangers. In summary, pathogen-associated molecules or immunologic adjuvants do not only ensure optimal activation of DCs to become professional APCs, but also control the activation status of naïve B cells with a BCR fitting for the prevalent antigens of the microbe presently challenging the immune system.

If optimal B cell activation is guaranteed, many copies of BCR/antigen complexes are delivered to phagosomes where they are processed, and peptides are generated from the recognized antigen. As B cells, besides of DCs and macrophages, express MHC II they can present the processed peptides on their surface. This happens in the protected environment of a lymph node (or the spleen), where previously naïve Th cells specific for the same antigen had also been activated by APCs and clonally expanded into many identical effector Th cells. Then—and only then—this B cell clone will receive the T cell help that it requires to become fully activated to start to proliferate and to clonally expand (Figure 1C). Proliferation of antigen-specific B cells and further differentiation to antibody-producing B cells, as well as memory B cells, takes place in a specialized compartment within the lymph node, known as the germinal center (reviewed in [64]). In summary, pathogen-associated molecules or immunologic adjuvants do not only ensure optimal activation of DCs to become professional APCs, but also control the activation status of naïve B cells with a BCR specific to the prevalent antigens of the microbe challenging the immune system.

### 2.5. B Effector Lymphocytes Have Several Options to Contribute to an Ongoing Immune Response

The most important product of activated effector B cells in an ongoing immune response to a microbial challenge are antibodies specific to antigens of the prevalent challenge. Therefore, after a first infection with a dangerous microbe, it is of utmost importance to have antigen-specific Abs in large numbers at one’s disposal as quickly as possible. Equally important is that the high-quality Abs with broad applicability are available for as long as the challenge persists. Clonal offspring of the antigen-specific B lymphocyte have several options in contributing to the ongoing adaptive immune response (reviewed in [65]) (summarized in Figure 2).

#### 2.5.1. Immediate IgM Production and Release: Increasing the Quality of Defense Mechanisms by Specifically Directing Attack Mechanisms to the Labelled Surface of Microbes

After the initial activation with T cell help (Figure 2A), some of the emerging offspring of the antigen-specific B cells develop into IgM-producing B cells. Following a brief proliferative phase, these B cells stop dividing and as plasma blasts immediately produce and release IgM antibodies within the lymph node (Figure 2A far left). While most of these B cells die after a few days, some develop into long-lived B cells that release antigen-specific IgM Abs for a prolonged time. In addition, a few of these B cells will receive a survival signal and develop into IgM-positive long-lived memory B cells (Figure 2A left) (reviewed in [66]). Due to the genetic organization of the loci encoding heavy and light chains of antibodies, the early first wave of antibodies will be of the IgM isotype. IgM antibodies bound to antigens on the surface of microbes are excellent activators of the complement system, but due to their large size of around 900 kDa, they circulate in the blood and are normally unable to penetrate into tissues. Upon re-encountering their antigen, memory B cells can be reactivated to start a second round of clonal expansion (not shown here for sake of clarity), again producing plasma blasts that immediately produce IgM antibodies specific for the antigen with the same affinity for the antigen as in the first encounter (Figure 2B left).

#### 2.5.2. Class Switching from IgM to IgG

The size of an Ab of the IgG isotype is around 150 kDa, which is much smaller than an IgM Ab of approximately 900 kDa. IgG molecules can cross the endothelial barrier with the help of a special receptor for IgG and enter the tissue, where they can move much faster than IgM to the site of injury or inflammation. Several different types of Fc receptors for IgG molecules exist on different types of leukocytes, increasing the versatility of effector functions (reviewed in [67]). In summary, switching classes from IgM to IgG is of great benefit for the immune system to increase not only the quantity of defense options (very large numbers of antigen-specific Abs), but also the quality of defense mechanisms (several types of Fc-gamma receptor-activated effector mechanisms). Finally, IgG Abs are of moderate to high affinity. By complex and tightly controlled processes called somatic hypermutation and affinity maturation, the affinity of IgG Abs can be increased while maintaining the antigen specificity (reviewed in [68]. Most of the clonal B cell offspring undergo several rounds of proliferation in the specialized compartment of the germinal centers (GC) of lymph nodes (Figure 2A middle). At this point in their development, these cells neither express antibodies (BCR) on their surface nor do they release antibodies. During this proliferative phase, individual B cells are allowed to switch classes of their antibodies. This means they re-arrange exactly those parts of their genes that encode the constant part of the heavy chain that defines the isotype of an individual Ab (reviewed in [69]). As this part of the Ab is not involved in antigen binding, the original antigen-specificity and the affinity for the antigen are retained during class switching. These B cells can develop into short-lived IgG antibody-producing B plasma blasts (Figure 2A middle). They also contribute to the first wave of antigen-specific Abs, now of the IgG isotype. As with the IgM-producing offspring, some of these class-switched B cells can also develop into long-lived IgG-producing B cells (plasma cells). In general, long-lived plasma cells are considered to be responsible for sustaining antigen-specific Ab titers for many months and possibly years after an encounter with a challenge (reviewed in [70,71]). And finally, a few of these class-switched B cells stop proliferation and receive a survival signal to become memory B cells, now expressing an IgG molecule as antigen receptor (BCR). These memory B cells do not contribute to the ongoing Ab production but remain dormant for many months or years. Upon re-encountering their antigen, they can be reactivated to start a second round of clonal expansion (not shown here for sake of clarity), producing plasma blasts that immediately produce antigen-specific IgG Abs with the same affinity as in the first round (Figure 2B middle).

#### 2.5.3. Increasing Affinity While Maintaining Specificity: Somatic Hypermutation and Affinity Maturation

Another group of activated B cells will go through rounds of proliferation within the germinal center. They will start a complicated Th cell-controlled process known as somatic hypermutation and affinity maturation to yield B cells with BCRs with the same specificity as the parent B cell clone, but with a higher affinity (Figure 2A right) (reviewed in [68]). The final result will be B cells that can produce antigen-specific Abs with a higher affinity than the first round of antibodies. Normally, such B cells will have undergone class switching as well, so these Abs of higher affinity will be of the more versatile IgG isotype. After a final specificity control by antigen-specific effector Th cells (not shown in Figure 2), a few of these B cells may release higher-affinity antigen-specific IgG Abs at the very late stage of the naïve immune reaction (Figure 2A far right). Some of these hypermutated and class-switched B cells will also receive a survival signal and become high-affinity IgG-positive memory B cells. Upon re-encountering their antigen, they can be reactivated in a very short time to start a second round of clonal expansion (not shown here for sake of clarity), again producing plasma blasts that immediately produce higher-affinity antigen-specific IgG Abs (Figure 2B right). This process can be repeated upon multiple recall situations until maximum affinity for the antigen is reached (not shown here). These mechanisms of creating immunological B cell memory are fundamental to vaccination procedures and have been studied in detail in mice and human beings (reviewed in [72]). It should be mentioned that the reactivation of memory B cells is regulated at several different levels. With respect to immunologic adjuvants, it is important to note that the extent of the initial activation of DCs dictates not only the whole activation process in the naïve situation but also controls the strength of antibody production and longevity of memory B cells (reviewed in [73,74,75,76,77,78]). In summary, optimally activated antigen-specific B cells have several options to contribute to an ongoing immune response (summarized in Figure 2).

#### 2.5.4. Antibodies Have a Short Half-Life and Need to Be Produced Continuously to Maintain an Effective Titer in the Blood

It is crucial to understand the difference between antibody titer and half-life of an independent antibody molecule. To protect us for a longer time, an effective titer of nAbs against critical antigens of a microbe has to be maintained for several weeks, ideally in blood and tissues. This makes sense as “ping-pong” reinfections within a population are highly likely. Protection can only be guaranteed if nAbs against a dangerous microbe circulating within a population are available throughout the critical infection period. Yet, once an individual Ab molecule is released from the producing B cell, it has a relatively short half-life of only a few days. The most relevant IgG isotypes usually have a half-life of around 3 weeks (reviewed in [67]), which is long compared to other isotypes, which only have a half-life of 2 to 6 days. As such, long-lived B plasma cells have to continuously produce large amounts of nAbs to maintain an individual Ab specificity. However, long-lived B plasma cells also do not live forever and are slowly lost within months or years, depending on the initial strength of the challenge (reviewed in [79]). How is it then possible to maintain sufficiently high nAb levels to achieve BoNT/A immunoresistance for several years? This boils down to immunological memory and the reactivation of memory cells.

### 2.6. Response to BoNT/A Injections in the Naïve Situation

#### 2.6.1. Activation of DCs by BoNT/A Injections

If a BoNT/A product containing components that may serve as immunologic adjuvants is injected, resident DCs may be activated. Different components in CPC-BoNT/A products can activate the immune system. Flagellin, reported to be contained in abobotulinumtoxinA [19,25], is a potent activator of human DCs by binding to TLR5 (reviewed in [80]). Flagellins are being developed as adjuvants in vaccines (reviewed in [81]). Bacterial DNA, reported to be contained in onabotulinumtoxinA [26], is a potent immune stimulator via TLR 9 (reviewed in [82,83]). Parts of bacterial DNA are employed as adjuvants for vaccines (reviewed in [84]). Clostridial hemagglutinins, also referred to as complexing proteins, have been identified as immune stimulators [85,86,87,88,89,90,91]. In addition, some BoNT/A products contain inactive toxin molecules [17]. Inactive proteins are frequently denatured or proteolytically cleaved and tend to form protein aggregates. Protein aggregates are a major cause for antibody formation against pharma proteins (reviewed in [92,93,94]) as they activate scavenger receptors on DCs.

It should also be mentioned that the procedure of injecting alone, irrespective of what is being injected, will result in the destruction of some tissue cells along the injection channel. Destruction of cells (necrosis) always results in the release of intracellular proteins, such as heat shock proteins, into the extracellular milieu. Being in the wrong compartment, these molecules serve as endogenous alarm mediators or damage associated molecular patterns (DAMPs). As resident macrophages and DCs express receptors for such DAMPs, they respond and become alerted (reviewed in [95,96]) to organize removal of necrotic material and initiate tissue repair. These alarm signals may contribute as one signal of many to the activation of DCs as these act as signal integrators (reviewed in [55,56]). This is irrespective of the presence of immune-stimulatory components in BoNT/A products.

DCs, optimally activated by one of these components or a combination of them, will ingest everything that is present in their immediate vicinity. This includes the injected 150 kDa BoNT/A molecule. However, in the absence of immunologic adjuvants, DCs will not be stimulated to perform phagocytosis, and a pure bioactive BoNT/A molecule should not be taken up. As an exotoxin that is released by the bacteria, it is not a prototypical ligand for PRRs and is most likely unable to activate DCs by itself. If there is no DC activation, peptides thereof will not be presented to T lymphocytes. This is a plausible explanation why injection of a pure and bioactive CPF-BoNT/A product does not induce antibody production, as has been observed in more than a decade of clinical practice in patients that were treated exclusively with CPF-INCO [28,29,30,31,32,33]. While there is no direct evidence demonstrating that injected BoNT/A is taken up by DCs, indirect data obtained during development of BoNT/A vaccines strongly suggest that BoNT/A is processed by this classic immunological pathway leading to antibody production [97].

#### 2.6.2. Presentation of BoNT/A-Derived Peptides to BoNT/A Peptide-Specific T Cells

Over 2 decades the seminal work of Atassi, et al. uncovered that BoNT/A is processed to peptides that are capable of inducing a specific T cell as well as an antibody response in mice and human beings (reviewed in [98,99]). T cell epitopes have been identified and characterized in BoNT/A heavy and light chains [100,101,102,103]. In patients treated with CPC-BoNT/A formulations for neurological disorders, the frequency of T lymphocytes specific for BoNT/A-derived peptides was investigated. A surprisingly high proportion of these patients (70%) showed T lymphocytes specific for BoNT/A-derived peptides, while untreated persons showed a very low background frequency of only 3% [104]. This result demonstrates that the T cell arm of the human adaptive immune system, if activated properly, responds to the bacterial antigen BoNT/A. Regrettably, such studies have not yet been performed with patients treated with a CPF BoNT/A product.

It was also demonstrated that human MHC II haplotypes influence T cell recognition of BoNT/A [105]. This in turn allows the conclusion that BoNT/A peptides have to be presented by MHC class II molecules. As it is now widely accepted that professional antigen presentation in the naïve immune response is by DCs (reviewed in [44,45]), it is logical to infer that injected BoNT/A is indeed taken up by DCs, processed to peptides that are presented via MHC II to elicit a T helper cell response.

#### 2.6.3. What Is Known About B Cell Responses to BoNT/A?

The frequency of patients responding to repeated injections with CPC-BoNT/A formulations by generating BoNT/A-specific B effector cells is not known. From several clinical studies and experience from more than 30 years, one can deduce that between 0% and 27% of patients respond by producing nAbs to BoNT/A (reviewed in [3,4,5,10]) that result in a decline in clinical efficacy. Interestingly, the higher frequency of nAb formation of over 20% was reported for “first” or “old” ONA formulation that contained a heavier load of bacterial components that could serve as immunologic adjuvants. After optimizing the purification protocol and introducing the “new” ONA formulation, a corresponding decrease in frequency of nAbs was noted (reviewed in [7]) [106,107]. However, one can reasonably assume that the frequency of B cell responses to CPC-BoNT/A formulations is likely to be higher compared to nAb formation. This is because only blood samples from clinically apparent secondary non-responders are tested and ascertained for the presence of nAbs to BoNT/A. However, it is likely that the proportion of non-neutralizing BoNT/A-specific antibodies is higher than that with nAbs [2,108]. Yet, the presence of non-neutralizing BoNT/A-specific antibodies has not been systematically investigated because they seem clinically irrelevant. Thus, the extent of the B cell response to CPC-BoNT/A currently remains unknown. One can argue that only nAbs are clinically relevant because they inhibit the uptake of the BoNT/A molecule into the nerve terminal. That said, non-neutralizing antibodies may also affect the behavior of injected BoNT/A molecules in the tissue. Simply by binding to BoNT/A, non-neutralizing Abs will increase the size of the neuromodulator molecule, and likely affect the speed of spread through the tissue to the nerve terminal as this depends on the size of the complex. In addition, non-nAbs of the IgG isotype will label BoNT/A molecules and make them available for the binding to receptors for immunoglobulins (Fc gamma receptors) on macrophages and DCs, possibly increasing the speed of active removal of the injected BoNT/A molecules.

#### 2.6.4. Role of BoNT/A-Specific IgM or IgG Antibodies

Presently, little is known on the role of IgM molecules in the immune response to injections of BoNT/A in patients and if IgM is relevant for developing immunoresistance. Early vaccination studies in human beings showed that IgM molecules specific for BoNT/A are generated after repeated injections [109,110,111] and are later followed by IgG isotypes. Principally, nAbs of the IgM isotype can be found in humans responding to BoNT/A [112], yet their contribution to immunoresistance remains unclear. Due to their large size, IgM molecules are circulating in blood and normally do not penetrate into tissues. In consequence, anti BoNT/A-specific IgM antibodies would not reside in the tissue where the pharmaceutical is injected and would not be available at the site of injection to bind to and neutralize BoNT/A before it reaches the nerve terminal. Therefore, it can be assumed that nAbs leading to SNR in patients should mainly be of the IgG isotype. Results from vaccination studies also suggest this [113].

## 3. How the Immune System Remembers—The Recall Situation

### 3.1. The Innate Immune Arm Does Not Contribute to Long-Lived Memory

After the first contact with a challenge, innate and adaptive arms of the immune system are activated. The innate immune system responds to the challenge immediately and engages very effective tools to combat the challenge. However, these tools are unspecific in nature, e.g., reactive oxygen species (ROS) or nitrogen oxide (NO), both being very potent anti-microbials that chemically modify (and inactivate) proteins and nucleic acids. However, this not only harms microbes but may also affect the cells producing these agents. To avoid mutations in nucleic acids that could result in neoplasia, all cells of the innate immune system die rapidly after activation. Thus, the innate immune system does not contribute to the immunological memory simply because dead cells do not remember. Consequently, after encountering the same dangerous microbe again, innate recall reactions are stereotypically identical to those in the naïve situation.

### 3.2. The Adaptive Arm of the Immune System Remembers

The human immune system can learn after successfully defeating a microbe. If it encounters a challenge such as a microbe that had been classified as dangerous in a previous fight again, the resulting antigen-specific response will be more rapid and efficient. Usually, the response will be so efficient that the persons affected will not become (severely) sick. Antibodies and memory T and memory B cells of the adaptive immune system are responsible for this antigen-specific memory.

### 3.3. Different Types of B Cells Are Responsible for Prolonged Antibody Production

Unlike a naïve situation where only a single antigen-specific T or B cell may be available, memory lymphocytes (memory T and memory B cells) are present in larger numbers in a recall situation due to prior clonal expansion. In addition, memory lymphocytes have a lower activation threshold compared to naïve T and B cells and can be rapidly activated outside of a lymph node and within tissues. Memory B cells also do not require the support of Th cells for activation. In principle, activation of such a memory B cell can be achieved by very high concentrations of the antigen alone (at least in vitro). This means that after encountering the same challenge again, memory B cells are able to produce large amounts of antigen-specific Abs, IgM, and IgG with moderate and higher affinities, practically immediately. In addition, these reactivated memory B cells can start to proliferate clonally and undergo a second or third round of somatic hypermutation and affinity maturation with the help of antigen-specific memory T helper lymphocytes. Consequently, large numbers of antigen-specific Abs are rapidly produced in a recall situation to levels achieved in the naïve situation, or even higher in the later phase of a recall situation. Finally, out of these pools of expanded B cells, new memory B cells will be generated in large numbers for a possible third or fourth encounter with this challenge in the future. This sequence of events is utilized in vaccination. It explains why injecting a vaccine repeatedly within a certain time frame can generate an antigen-specific protection for a very long time (reviewed in [114]).

### 3.4. Reactivation of Memory B Cells Is More Effective in the Presence of Immunologic Adjuvants

Experiments performed with isolated memory B cells in vitro and in mice models demonstrated that reactivation of dormant memory B cells was possible by high concentrations of pure antigen, especially if these antigens were provided in a multivalent form, i.e., in larger aggregates with repetitive epitopes. This is probably due to the ability of such antigens to effectively crosslink many BCRs on the surface of memory B cells that have a lower activation threshold per se. This knowledge has been used to develop vaccines, e.g., by crosslinking smaller antigens with formaldehyde into larger complexes. However, more than 100 years of experience with vaccines showed that activation and reactivation of immune cells is much more effectively achieved in the presence of immunologic adjuvants (reviewed in [115]). It has also become clear that the strength and duration of immunological memory are influenced by the presence of substances that can act as immune stimulators or immunologic adjuvants by binding to pattern-recognition receptors [116,117]. It was initially thought that this was only due to the optimal activation of DCs and naïve B cells at the beginning of an immune response, but it soon became clear that pattern-recognition receptors are also expressed on memory B cells [118]. While in vitro (polyclonal) activation of B cell memory was possible by TLR agonists alone, activation of BCR by antigen was also necessary in vivo, which resembles the real-life situation [119]. Both types of signals were required in vivo to activate memory B cells (reviewed in [120]), and signaling via pattern-recognition receptors appears to amplify the activation of memory B cells [121,122].

In summary, memory is much better reactivated by antigen in combination with immunologic adjuvants than by pure antigen alone in a real-life situation. From an infection point of view, this makes sense as it is highly unlikely that a person will be challenged by a pure antigen alone in nature. Instead, a more realistic scenario is re-infection with an intact microbe consisting of proteins that serve as antigens, in combination with microbial surface structures or nucleic acids that can function as danger signals or adjuvants.

### 3.5. Immunological Memory and Repeated Injections of BoNT/A—Role of Immunologic Adjuvants

However, this may be different in a clinical setting as it is possible to inject pure antigen in the form of a CPF-BoNT/A product. Conversely, injecting a CPC-BoNT/A product will introduce the antigen with potential immune stimulatory components that may serve as immunologic adjuvants. As BoNT/A is injected repeatedly over a long time, this type of treatment closely resembles a vaccination. Therefore, the underlying principles we have described for an immune response in general will apply to BoNT/A therapy in both naïve and recall situations. The outcome in a recall situation may well be dependent on the purity of BoNT/A product injected and its ability to reactivate BoNT/A-specific memory B cells.

If patients are treated with CPC-BoNT/A product over a long time, their immune system may become activated, resulting in a persistently high titer of nAbs. This will then lead to partial or complete immunoresistance. This high level of nAbs is maintained because every time the antigen (BoNT/A) is injected anew with immunologic adjuvants, a new reactivation of BoNT/A-specific memory B cells occurs. Together with long-lived nAb-producing plasma cells, these reactivated memory B cells and their offspring release nAbs continuously for months and years, and this cycle of reactivation is repeated with every regular BoNT/A injection. This explains how a high level of nAbs is maintained although each individual antibody molecule has a relatively short half-life of about 3 weeks (for IgG).

## 4. Case Report

### 4.1. Clinical History

We hereby present a real-life case example to illustrate the immunological concepts described above and demonstrate how the absence of immunologic adjuvants can prevent the reactivation of immune memory. A 47-year-old Chinese female with no known medical history had been receiving BoNT/A treatment for masseter reduction for 4 years between 2015–2019, on average once every 6 months. Based on her memory and account, the BoNT/A product used previously had always been ONA. The dose was reported to be 40–50 units per masseter per session. Since 2018, the patient had noted that her treatments were getting less effective with increasingly shorter duration of effect and requiring more frequent treatments (interval creep). In January 2019, she received 50 units of ONA to each masseter but did not experience any clinical effect. As such, she presented to one of the authors’ clinic (Radium Medical Aesthetics, Singapore) in February 2019 to seek a second opinion. On examination, her masseters were large and palpable despite the high dose of ONA received. She was injected again with 34 units of INCO to each masseter. The treatment still yielded no clinical results, and she was suspected to have developed immunoresistance to BoNT/A. The patient’s serum was subsequently sent to a specialized laboratory (toxogen GmbH, since January 2023 toxologics GmbH, Hannover, Germany) in August 2019 for an ex-vivo mouse hemidiaphragm assay (MHDA), which confirmed the presence of nAbs to BoNT/A and clinical diagnosis of nAb-induced complete SNR. In October 2019, the patient’s serum was obtained again to specifically measure nAb titer via MHDA. The highly sensitive MHDA’s cutoff point for nAb detection is 1.82 mIU/mL and up to a maximum of 12.15 mIU/mL [4,36,41] (A.Rummel personal communication). The patient’s nAb titre returned as 5 mIU/mL.

As this patient was unwilling to discontinue BoNT/A treatment completely in the hope that she might become responsive again at some point in time, she continued to receive INCO every 3–4 months, from July 2019–April 2024, with a dose of 50 units per masseter (total 100 units). This approach allowed the patient’s nAb titers to be monitored closely over time while receiving continuous CPF-INCO. It should be noted here that in general, the units of different BoNT/A products are not interchangeable (see FDA approvals of INCO and ONA [20,21]) (reviewed in [123,124,125]). At every visit, photographs were taken to monitor clinical response. Her serum was obtained and sent for MHDA, followed by injection of INCO into both masseters. The main objective was to monitor her trend of nAb titers over time with continuous INCO injections.

The patient’s nAb titers were progressively on a downward trend from July 2019, eventually reaching below the MHDA’s lower cutoff point of 1.82 mIU/mL in December 2022 (Figure 3) and was hence undetectable.

However, clinical response assessed as a visible reduction in the size of the masseters on clinical photography was only observed from January 2024 onwards. The patient began to show mild clinical response from January 2024, with more obvious results in May 2024 (Figure 4A). This was also shown quantitatively on clinical ultrasonography with reduction in masseter muscle thickness bilaterally (Figure 4B).

### 4.2. Summary of Case Report

This report demonstrates the first known case of complete SNR due to nAb formation from aesthetic BoNT/A use in Singapore. It is also the first case study documenting a decline in nAb titers despite continuous and regular INCO injections in aesthetic practice. Following diagnosis of complete SNR, the patient continued to receive CPF-INCO at regular intervals despite being non-responsive to monitor the trend of nAb titers and return of BoNT/A effects. She began to only show signs of clinical response after 3 years when nAbs were no longer detectable. The patient’s clinical response to BoNT/A correlated with nAb titers, but there was a period of delay between the resolution of nAb titers and the return of clinical response. The patient only demonstrated clinical response about a year after nAbs have become undetectable. Clinical response was seen sooner in the glabella (Figure 4C) from September 2023 onwards compared to the masseters (Figure 4A). This could be due to the difference in size of the muscles—significantly more neuromuscular junctions need to be blocked in the masseters compared to the glabella for visible results. Despite continuous treatment with CPF-INCO from July 2019–April 2024, BoNT/A antibody titers continued to fall, demonstrating that CPF-INCO did not in this case contribute to the formation of de novo BoNT/A antibodies.

## 5. Discussion

Physicians have two different types of BoNT/A products to choose from to treat their patients: those that are free of complexing proteins including CPF-INCO [21], daxibotulinumtoxinA [126], Coretox^®^ [127], and most recently relabotulinumtoxinA [128,129], and those that contain them (CPC-ONA or CPC-ABO and many others). Exclusive treatment with CPF-INCO did not lead to nAb production in the past presumably because immunologic adjuvants pivotal for the activation of DCs and naïve B lymphocytes in the naïve situation were missing. In consequence, patients treated exclusively with CPF-INCO to date did not develop immunoresistance [28,29,30,31] even at very high doses [28], notwithstanding the relatively short observation periods of 12–16 weeks. CPC-BoNT/A products contain the core neurotoxin in a large complex together with complexing proteins and frequently other bacterial components that have the potential to serve as immunologic adjuvants. These CPC BoNT/A products induce the formation of nAbs to BoNT/A in patients with a low but clinically relevant frequency in clinical and aesthetic indications (reviewed in [3,4,5,6,7,8,9,10,11]). Once nAbs titers reach a critical level, these patients become partial or even complete secondary non-responders. This limits the use of BoNT/A for further treatments due to loss of efficacy or reduction in duration of the desired effect. Continuation of treatment in such non-responders with CPC-BoNT/A products will maintain a high titer of nAbs throughout the treatment because memory B cells are permanently reactivated by the combination of the antigen BoNT/A plus immunologic adjuvants. After stopping treatment, it will require several years before nAb titers subside to a clinically irrelevant level [14,36]. This relatively long time is most likely due to the gradual loss of BoNT/A-specific long-lived antibody-producing B plasma cells. There will not be any clinical effect as long as nAb titers are high enough to neutralize all BoNT/A molecules, regardless, whether a CPC- or CPF-BoNT/A product is injected. When nAb titers reach a level that is lower than the amount of injected BoNT/A molecules, the patient will become responsive to BoNT/A again—first partially, then completely when all nAbs have disappeared. If treatment is re-started with a CPC-BoNT/A product, there is a risk of dormant memory B cells being re-activated to produce new nAbs, and their offspring will develop into short- and long-lived Ab-producing B cells, as well as a new generation of dormant memory B cells. This may happen even after many years of treatment holiday and resolution of nAb titers [35]. However, if treatment is restarted with CPF-INCO [35,36], we propose that the activation signals will not be strong enough to re-activate dormant memory B cells due to the absence of immunologic adjuvants. This novel approach to manage nAb-mediated SNR with a low immunogenic BoNT/A product seems to be quite attractive for some patients, but it has limitations.

One limitation of this approach is that it is impossible to predict which patient with partial immunoresistance will respond. Continuation of treatment with low immunogenic BoNT/A does not work in all patients, as evident in both studies by Hefter, et al. How can that be explained? Why do not all patients respond equally well? Inter-individual differences in immune response exist, and many possible reasons have been proposed (reviewed in [130]). It is beyond the scope of this review to discuss the reasons for these differences in detail. Yet, we all know from daily experience that some persons cope very well with a certain type of infection while others become severely sick. Amongst the many possible reasons, genetics play an important role. One important genetic aspect is that each person has an individual MHC (also known as HLA in humans) haplotype (for details see [131]). MHC is polygenic and highly polymorphic. MHC genes are expressed in a co-dominant fashion, so that each individual expresses both parental alleles, with a maximum of six different MHC class I and eight different MHC class II molecules, all of which can present peptides. However, the number of peptides that can be presented on each MHC molecule is limited by the amino acid sequence. Thus, inter-individual differences exist in the ability of APCs to present peptides to naïve T lymphocytes, a process mandatory for initiating an immune response leading to Ab production [131]. This seems to be applicable to BoNT/A, or more specifically, the peptides that can be derived from BoNT/A heavy or light chains. One study investigating the influence of certain MHC haplotypes on the recognition of BoNT/A-derived peptides by T lymphocytes found that HLA DQA*01:02 tended to be more effective in presenting specific BoNT/A peptides than HLA DQA*01:01. [105], suggesting that there is a possible genetic predisposition to producing nAbs against BoNT/A. It is also plausible that subtle differences in the ability to present BoNT/A-derived peptides may result in differences in strength and duration of adaptive responses, including antibody production, which partly explains the observed inter-individual differences. However, this currently remains hypothetical and awaits experimental proof. Differential expression and genetic predisposition of PRRs such as TLRs may also affect how an individual person responds to an immunologic adjuvant. While there is presently no published data with respect to BoNT/A, studies on allergies have demonstrated that genetic variations of TLRs can contribute to allergic diseases [132].

A second limitation is that is impossible to predict how long it will take for nAb titers to decline to a point where they are clinically irrelevant in an individual. Large inter-individual differences were reported with respect to persistence of nAb titers in BoNT/A patients before they dropped to an undetectable level [14,35,36,41]. Studies addressing the persistence of long-lived B plasma cells and memory B lymphocytes specific for BoNT/A are not available. Yet, it is reasonable to draw parallels to the extensive experience from vaccinations and infer that the longevity of these memory cells will vary from person to person. An important aspect for the long duration of such an antigen-specific response seems to be how strong the initial activation of the DCs was (reviewed in [73,74,75,76,77,78]). It is also tempting to speculate that the immune status of the person at the time of injections plays a role. Differences in activation thresholds of naïve and activated immune cells may exist due to other immune reactions running in parallel in a patient.

Given these limitations, it seems unrealistic and impractical to prophylactically continue BoNT/A treatment with a low-immunogenic product in all patients with nAb-mediated SNR, especially those with complete SNR, as this translates to unnecessary injections, high costs, and potentially prolonging the period of non-responsiveness. Patients with complete SNR will not experience any improvement in symptoms regardless of BoNT/A formulation until nAb titers have started to decline. As demonstrated by Hefter, et al. [36], some patients with partial SNR initially experienced a transient increase in nAb titers while receiving CPF-INCO, most likely due to a boostering of memory B cells despite the lack of immunologic adjuvants. This may be due to a lower activation threshold or a longer half-life of memory B cells in these patients. In such a situation, continuation of treatment with CPF-INCO might prolong the period of non-responsiveness rather than shorten it as compared to immediate cessation of treatment. nAb titers did not drop in 6 out of 37 patients over the study period of more than 4 years [36]. In these patients, “prophylactic” continuation of BoNT/A treatment seems hardly justified.

In the absence of reliable clinical parameters that can be utilized to predict which individual would benefit from this approach, is this purely an academic exercise? It all boils down to the lack of predictability of when is the best time point to inject a low-immunogenic BoNT/A product into an individual who has developed nAb-related SNR. This demonstrates the need for robust and simple-to-use diagnostic procedures to identify the ideal time point.

The ex vivo MHDA is extremely sensitive in detecting nAbs to BoNT/A (reviewed in [4,133,134] This assay has demonstrated a very high sensitivity compared to other detection methods, e.g., mouse protection assay (reviewed in [4,38,135]), but it also might carry the risk of a higher false-positive rate due to the possibility that low nAb titers of unclear clinical relevance may be detected. It is also resource and time consuming, and presently only performed in a few specialized laboratories, making it impractical as a first-line diagnostic test. An ideal test would be a stick assay that allows physicians to conveniently detect anti-BoNT/A Abs, both non-neutralizing and neutralizing Abs, in their clinical practice. A small sample of blood would have to be drawn still, but then results would be quickly obtained without the need for special technical equipment. This could also be achieved using an enzyme-linked immune sorbent assay (ELISA) specific for anti-BoNT/A Abs. Such ELISAs have been used and described [136,137,138]. A major limitation of a stick assay and/or an ELISA, in contrast to the MHDA, is their inability to discriminate between non-neutralizing and neutralizing anti-BoNT/A Abs. However, these structural assays would suffice as a first-line screening test to detect the presence of Abs to BoNT/A. Upon the identification of BoNT/A Abs, MHDA serves as a critical second step to distinguish nAbs from non-neutralizing Abs. It should be noted here that several studies have reported that a positive detection of nAbs does not necessarily correlate or predict SNR [7,41,139,140,141]. However, the positive detection of nAbs on MHDA, when correlated clinically with signs and symptoms of SNR, will allow physicians to confirm the diagnosis of nAb-mediated SNR.

## 6. Summary and Conclusions

Here, we present a scientific reasoning for the paradoxical phenomenon of declining nAb titers despite continuous injection with CPF-INCO that is observed in some but not all neurology patients. We’ve highlighted how immunologic adjuvants such as complexing proteins and other bacterial components contained in CPC-BoNT/A products are required for two essential steps leading to nAb production against BoNT/A in the naïve situation. In addition, we called to attention that memory B lymphocytes carry receptors for immunologic adjuvants and are more easily activated in a recall situation by a combination of antigen plus immunologic adjuvants than by the antigen alone. This explains why memory B lymphocytes specific for BoNT/A are much less likely to be reactivated by a pure BoNT/A formulation like CPF-INCO due to the lack of immunologic adjuvants. Thus, immunologic adjuvants come into action at least at three critical steps in the way to continuous and long-lasting antibody production. This explains why continuation of treatment with CPF-INCO may be a viable treatment option for some patients in the presence of nAbs.

Presently, however, the lack of predictability of when is the best time point to inject a low-immunogenic BoNT/A product into which patient without further boostering the BoNT/A-specific immune response limits the practical application of this approach. Novel diagnostic tools are required to identify promising candidate patients, and future studies with larger numbers of patients are needed to establish the average time frame required for the resolution of nAb titers and the return of clinical response to BoNT/A. As such, prevention of nAb formation by using a CPF-BoNT/A product with the lowest immunogenicity seems to be a more prudent choice to avoid immunoresistance in the first place.

## Figures and Tables

**Figure 1 toxins-16-00422-f001:**
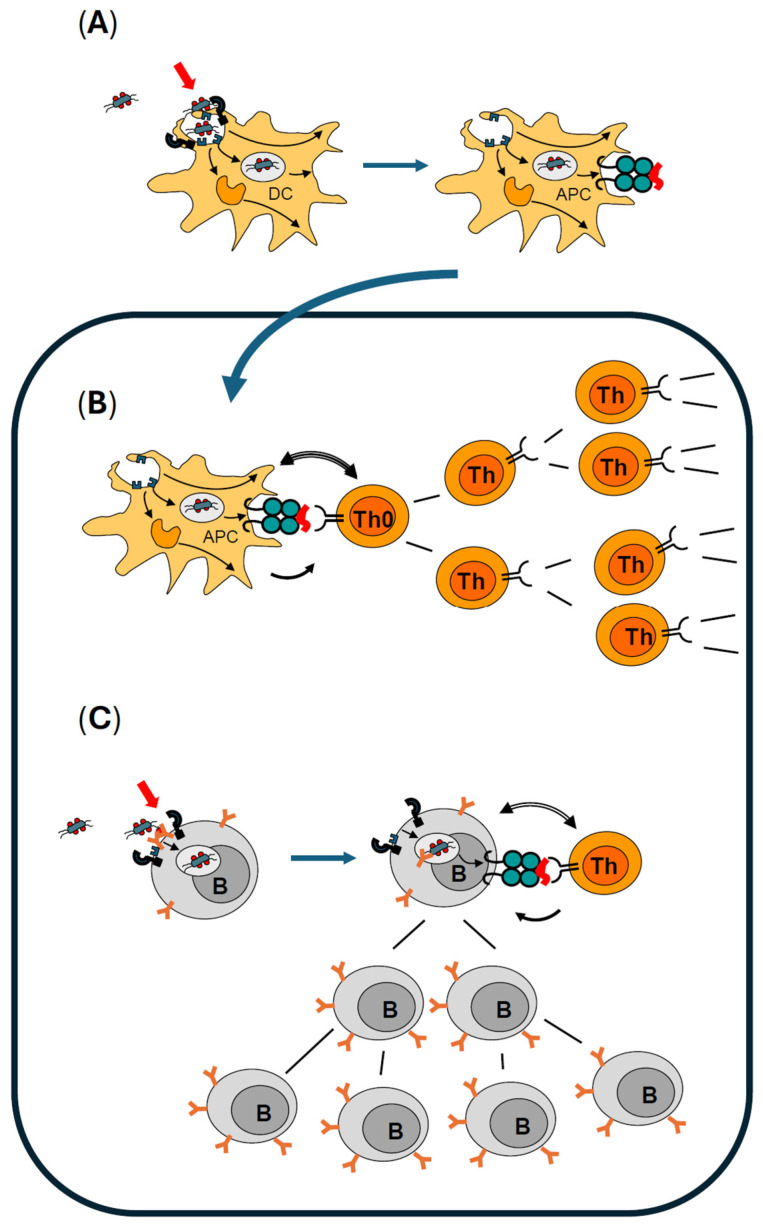
Summary scheme of the activation of the immune system: The naïve situation. (**A**) Dendritic cells (DCs) are sentinel cells residing in tissues. Upon encountering a microbial challenge, microbial surface structures, e.g., flagellin, act as immunologic adjuvants (red arrow) to activate pattern-recognition receptors (PRRs) on DCs. Activated DCs phagocytose what they have recognized (antigens) in their vicinity. Phagocytosed proteins are digested in phagosomes to antigenic peptides that are loaded onto MHC II. Upon optimal activation, DC move from the tissue into the next draining lymph node (rolling blue arrow from **A** to **B**). (**B**) In the lymph node, DCs settle as professional antigen presenting cells (APC) presenting peptides on MHC II to antigen-specific naïve T helper lymphocytes (Th0). If a naïve Th0 cell recognizes the presented peptide in MHC II, it interacts with the costimulatory surface molecules on the APC (double arrow) and receives cytokines from the APC (single arrow). The Th0 cell becomes fully activated, starts to proliferate, and clonally expands to a large number of effector Th cells with identical peptide specificity. (**C**) An antigen-specific naïve B lymphocyte (B) recognizes the same antigen with its B cell antigen receptor (BCR) that is a plasma membrane-anchored immunoglobulin. In the presence of immunologic adjuvants (red arrow) these PRR are engaged, and this B cell becomes optimally activated (left). It internalizes its BCR with the bound antigen, processes it to antigenic peptides (right), and presents these in MHC II on the surface (the same peptide as presented by the DC). One of the offsprings of the clonally expanded antigen-specific Th provides help in form of costimulatory molecules (double-headed arrow) and T helper cell cytokines (single arrow). This allows clonal expansion of this antigen-specific B cell to develop into many B plasma cells with the same antigen specificity. In the naïve immune response, immunologic adjuvants (red arrows) come into action twice: first by facilitating full activation of DC to become APC, and second, by allowing full activation of an antigen-specific B cell.

**Figure 2 toxins-16-00422-f002:**
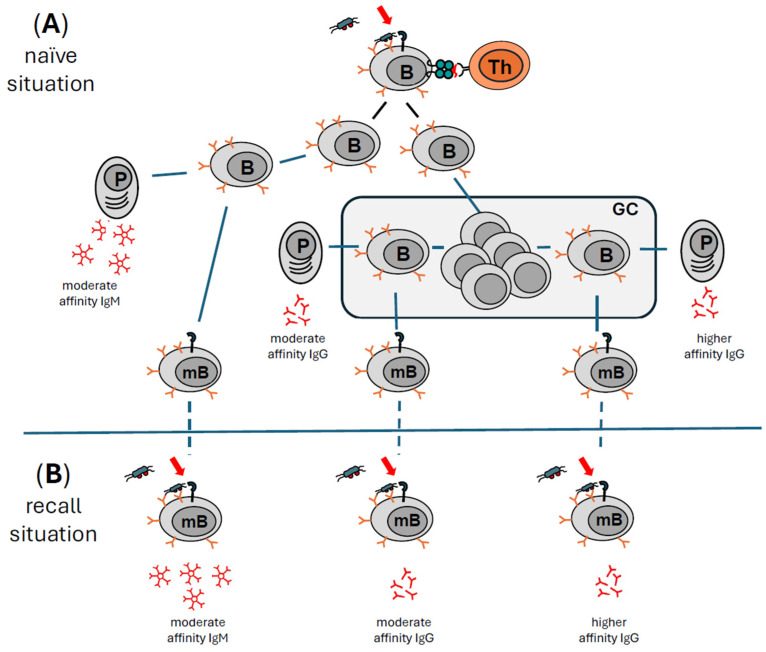
Antigen-specific B lymphocytes have several options of contributing antibodies in a naïve and memory situation. (**A**) The naïve situation. An antigen-specific B lymphocyte that has been optimally stimulated by its antigen plus immunologic adjuvants presents peptides to an antigen-specific T helper lymphocyte in the lymph node and receives T cell help to clonally expand. The clonal offspring have several options. They can develop into IgM-producing antibody factories, or plasma blasts (P) providing the first wave of protective antibodies with a low to moderate affinity (far left). A few of these IgM-positive B cells may develop into resting memory cells (left). In the germinal center (GC), B cells undergo several rounds of proliferation (middle). They can switch antibody classes and contribute with IgG antibodies with a similar affinity for the antigen to the first wave of Abs (middle). Some of these B cells can be rescued to become IgG-positive memory B cells (middle). Others can switch antibody classes and improve the affinity of their BCR by undergoing somatic hypermutation and affinity maturation. A part of these B cells can either become IgG-positive memory B cells (right) or produce higher-affinity IgG Abs in the late stage of an immune response (far right). (**B**) The recall situation. After encountering the same antigen again, all depicted types of memory B cells can rapidly produce antigen-specific Abs of different isotypes and affinities. Immunologic adjuvants can contribute to optimal B cell activation in the naïve and the recall situation (indicated by the red arrows).

**Figure 3 toxins-16-00422-f003:**
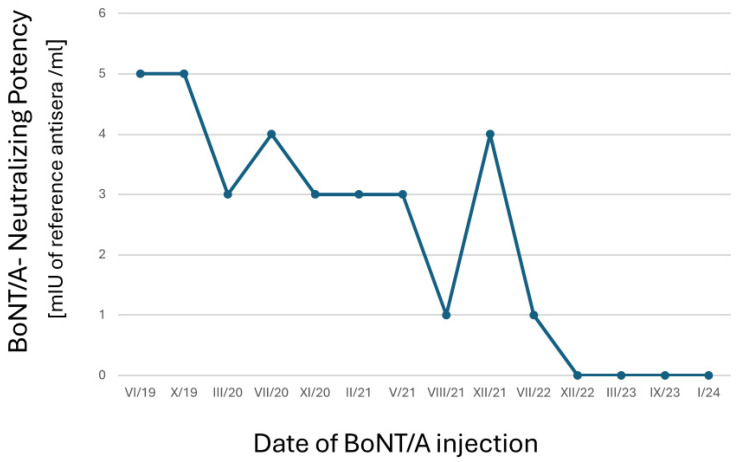
Trend of nAb titers over time. From July 2019 (VII/19) to January 2024 (I/20), the patient’s serum was analyzed for nAbs to BoNT/A using an ex-vivo mouse hemidiaphragm assay (service performed by toxogen GmbH, since Jan 2023 by toxologics GmbH, Hannover, Germany) at regular intervals. The patient received INCO every 3–4 months, from July 2019 (VI/19)–April 2024 (IV/24), with a dose of 50 units per masseter. Titers of nAbs were progressively on a downward trend from July 2019 (VI/19), eventually reaching below the lower cutoff point of 1.82 mIU/mL and hence regarded as undetectable in January 2023. (International unit (IU)/mL is a measurement of neutralizing BoNT/A activity in serum. One IU neutralizes 10,000 LD_50_ BoNT/A. The botulinum neurotoxin serotype A antitoxin standard was trivalent Botulismus Antitoxin Behring (registration no. 31a/78) Batch 080031A from Novartis Vaccines and Diagnostics GmbH & Co. KG, 35006 Marburg, Germany).

**Figure 4 toxins-16-00422-f004:**
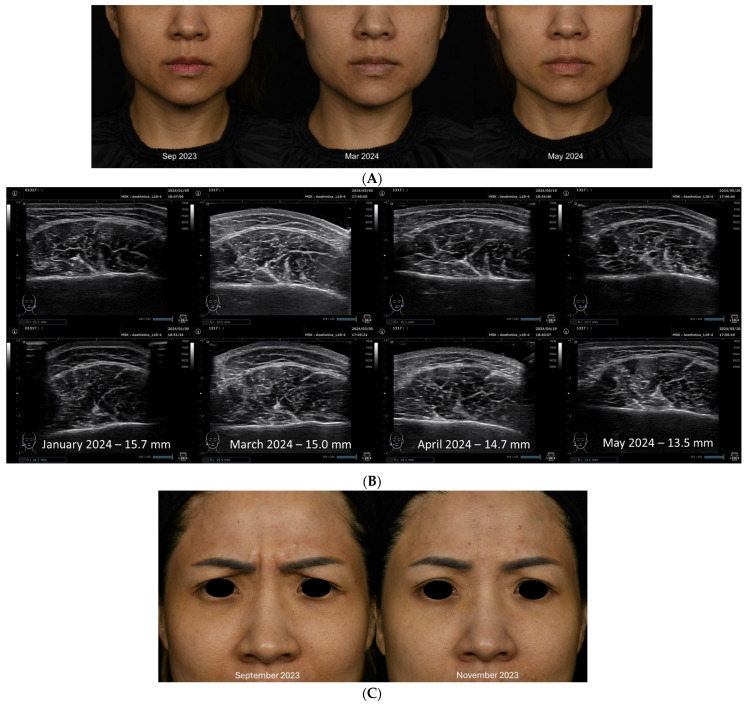
Assessment of clinical responses of patient. (**A**) Photographic demonstration of masseter reduction. Before incobtulinumtoxinA injection in September 2023 and March 2024; 2 months post-injection with INCO in May 2024. (**B**) Ultrasonographic measurement of masseter thickness reduction. Following INCO injection in January 2024, mean thickness of masseter decreased from 15.7 mm to 15.0 mm in March 2024, and from 14.7 mm in April 2024 to 13.5 mm in May 2024. (**C**) Photographic demonstration of clinical response to INCO treatment in the glabella at maximum frown: (left) before; September 2023 (right) after; November 2023).

## Data Availability

Not applicable.

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
