# Peer review of "Continuous Treatment with IncobotulinumtoxinA Despite Presence of BoNT/A Neutralizing Antibodies: Immunological Hypothesis and a Case Report"

_toxins, 2024, doi:10.3390/toxins16100422_

Round 1
Reviewer 1 Report
Comments and Suggestions for Authors
This manuscript discusses an important potential issue of chronic treatment with protein therapeutics. The article lays out basic concepts of immunology relevant to understanding the body’s immune response when exposed to foreign proteins. However, the intertwining of this to BoNT/A responses, particularly how “accessory proteins” act as adjuvants, is hypothetical as limited data/evidence is cited to support these statements. This reviewer suggests that supporting peer-reviewed literature be cited and that the language be amended to indicate the hypothetical nature of the narrative presented in this review.
- The title could reflect the hypothetical or proposed nature of the author’s perspective.
- Although many of the principles of immunology are articulated, they are often applied to toxin biology as if they have been proven, without providing evidence for the validity of their assertions. Indeed, the details of the mechanism of formation of neutralizing antibodies (NAbs) to minute quantities of purified commercial-grade neurotoxin is not established. The FDA has explicitly noted in the botulinum toxin package inserts that neurotoxin immunogenicity is complex. Notably, from the incobotA package insert, incobotA is not immune-inert.
- The authors should include a Limitations section which is more extensive, and highlights the hypothetical aspects of their proposal. This approach would provide scientific balance to their theories. Among the various limitations, the authors should note that notwithstanding their hypothesis, incobotA is susceptible to antibody formation (see incobotA package insert).
- Misuse of nomenclature:
o The authors inaccurately describe the 150 kDa “core neurotoxin” is the “active [pharmaceutical] ingredient”. The FDA has adopted the USAN-designated non-proprietary name to the Active Pharmaceutical Ingredient (see Karet GB (2019) How Do Drugs Get Named? AMA.J.Ethics 21:E686-E696. NLM: PM:31397664). Consequently, the full complexed toxin is the API.
o The authors claim that, in the setting of a protein complex, the associated proteins are “adjuvants.” This contradicts the current guidelines for describing protein complexes (see https://www.ncbi.nlm.nih.gov/genbank/internatprot_nomenguide/; Last updated: 02-MAR-2020). Use the “adjuvant” terminology to describe complexing proteins is not established. Notably, it is highly unlikely that regulatory agencies would approve “a BoNT/A product that contains adjuvants” (line 232). Throughout the article, the authors utilize the word liberally.
o The authors further classify the complexing proteins as “unnecessary”. This assertion is not established. They suggest that the complexing proteins are not “bioactive”. Challenges to this approach were recently reviewed by Avelar R (2024) Botulinum Toxin Accessory Proteins: Are They Just an Accessory? Dermatol.Surg. NLM: PM:38864825.
- Line 75: Use of the word “many” is inappropriate as the references included total to 4 with limited primary data (non-review, peer-reviewed publications); “several” would be a more appropriate adjective.
- Lines 138-141: Suggest refrain from using condescending language.
- Lines 161- 170: Include a reference for these statements.
- Figure 1: This figure is very confusing and difficult to follow; several details in this figure are hypothetical. Those proven should be referenced. For example:
o Lines 216-217: Please provide reference of data showing BoNT/A is phagocytosed by DCs.
o Lines 218-225: Need references to show that MHC class II presents BoNT/A antigen and that there is clonal expansion (Th and B cell proliferation) is response to BoNT/A exposure.
o Line 225: Has B cell phagocytosis of BoNT/A been demonstrated? Please provide reference.
o Line 230: Please also provide reference that adjuvants amplify clonal expansion
- Line 235: Need reference
- Lines 237-238: Need reference
- Line 255: what are the conditions needed to generate 10E3+ clonal offspring, and has that been demonstrated in routine patients receiving BoNT therapy?
- Lines 293ff: difficult to reconcile with the figure
- Figure 2 is mentioned in the body of the article as a general immunological concept, but referred to as BoNT/A- specific in the legend of the figure. If this is BoNT/A specific, references are needed to support each statement.
- Line 454: this statement should include the concept that memory B cells can be reactivated only above a certain threshold and within a certain timeframe, both of which vary depending on the amount and nature of the initial exposure.
- Lines 463, 478 & 542: the authors discuss “dangerous microbe”, “dangerous challenge” and “danger signals or adjuvants” and make the association to BoNT/A. Rather than comparing a licensed pharmaceutical to a “dangerous” substance, the more relevant and less inflammatory wording might be “foreign protein”.
- Line 516: Please provide data to support the assertion that CPC products stimulate immunological memory more effectively than CPF-INCO?
- Case report:
o The authors should note that incobotA units are not the same as onabotA Units. The FDA label and additional published data have demonstrated that the Units of onabotA can not be converted into units of incobotA (see Xeomin package insert: “WARNING: Dosing Units of botulinum toxins are not interchangeable between commercial products.” This is supported peer-reviewed literature.
o Notably, the patient was exposed to ~40U onabotA per masseter per session (total 80U). The patient was subsequently treated with a total of 50U incobotA. The treatment wore off prior to expected, and the authors concluded that the patient was a secondary non-responder. Notwithstanding non-interchangeablity, this conclusion is not clinically supported since a lower dose of incobotA would be expected to result in a less robust response compared to the administered dose of onabotA. Indeed, when a higher dose of 73U incobotA was administered, the expected response was achieved.
o Figure 3:
§ The X axis legend is not interpretable
§ The graph should be annotated with dates of treatment and dose administered
o Figure 4:
§ it would be helpful to annotate the figures with the dates as opposed to listing in the legend
§ Are the figures at maximum contraction?
§ How long after each injection were the figures taken?
o What was the clinical indication to continue to treat the patient when they were non-responsive to incobotA?
o The authors state that incobotA “does not contribute to” the formation of antibodies. However, the incobotA Package Insert has examples of nAbs. Please explain this contradiction.
- Figure 5:
o This image is hypothetical and should be indicated as such. If this graph represents data from conducted studies, please provide the summarized data points in the article or provide the reference.
o I do not think the author’s theory summarized in the figure explains the nAb titer response for one of the three groups of patients where there was an observation of a transient increase in nAb titer following switch to CPF-INCO in the first two years before falling back down again (see text on page 3/27 of the manuscript). In this group of patients, they describe a booster phenomenon with the identical antigen but in the other two groups, they hypothesize the importance of adjuvants and its dual signaling on B-cells. Furthermore, if it acts as a booster, then why would the nAb titers fall after two years? This would be a limitation of the hypothesis.
o Also, it was unclear why the relative clinical response was higher on the right vertical axis, but NAbs were higher on the left vertical axis. Graphically, I would anticipate the higher response on the right pointing down, consistent with the lower nAb titre
- Line 713-714: “…since the active neurotoxin is the same in all commercially available formulations…” see comments above regarding nomenclature. In addition, the tertiary structure and post-translatable changes to these complex proteins may not be identical, notwithstanding that the amino acid sequence may be the same.
- Line 721: Memory B cells and adjuvants were not specifically studied in these references.
- Line 723: Please either provide reference or state that this is the author’s hypothetical assessment regarding the threshold of number of molecules of BoNTA. Is the concept based on molar concentration of the total injected molecules?
- Line 726: In the section of limitations, the authors need to address under what circumstances to utilize their approach of continued treatment, and, what potential unintended medical consequences may occur. Measurements of neutralizing antibody titres are not generally available, and therefore the author’s approach is not practical. The approach of continuing incobotA in a patient who is a secondary non-responder can lead to further enhancement of the immune state.
Author Response
Response to Reviewer 1 Comments
Author's Reply to the Review Report (Reviewer 1)
First, we thank the reviewer spending so much time and effort in carefully reviewing our manuscript, and for the many helpful suggestions to enhance the clarity of the work and make it more useful and appealing to readers. All suggestions and feedback were taken into account and duly considered in the amended version of this manuscript.
As the reviewers have requested for major revisions, we had to reorganize the text and add additional information and references, which resulted in some restructuring of the manuscript. Where possible, we will refer the reviewers to the new position the respective points in the revised version.
Please find our detailed point-by-point responses below, with the reviewer’s comments italicized and bolded for clarity.
Point-by-point Response to Comments and Suggestions for Authors
1) This manuscript discusses an important potential issue of chronic treatment with protein therapeutics. The article lays out basic concepts of immunology relevant to understanding the body’s immune response when exposed to foreign proteins. However, the intertwining of this to BoNT/A responses, particularly how “accessory proteins” act as adjuvants, is hypothetical as limited data/evidence is cited to support these statements. This reviewer suggests that supporting peer-reviewed literature be cited and that the language be amended to indicate the hypothetical nature of the narrative presented in this review.
We thank the reviewer for this comment. The reviewer has rightly pointed out the hypothetical nature of some our discussion that intertwines the basic concepts of immunology and their application to BoNT/A responses. We have now separated these two aspects completely by introducing new sections “2.6. Response to BoNT/A injections in the naïve situation” and “3.5. Immunological memory and repeated injections of BoNT/A – role of immunologic adjuvants”. We’ve also clearly stated where the interpretations were of a hypothetical nature, and cited more peer-reviewed literature supporting our BoNT/A-related interpretations. Figures 1 and 2 have been edited to show only basic immunology concepts, and all BoNT/A-specific illustrations have been removed. BoNT/A-specific aspects are discussed in the new paragraphs 2.6. and. 3.5.
2) The title could reflect the hypothetical or proposed nature of the author’s perspective.
We thank the reviewer for this comment and have revised the article title to “Continuous treatment with IncobotulinumtoxinA despite presence of BoNT/A neutralizing antibodies: Immunological hypothesis and a case report” to clearly reflect the hypothetical nature of our discussion.
3) Although many of the principles of immunology are articulated, they are often applied to toxin biology as if they have been proven, without providing evidence for the validity of their assertions. Indeed, the details of the mechanism of formation of neutralizing antibodies (NAbs) to minute quantities of purified commercial-grade neurotoxin is not established. The FDA has explicitly noted in the botulinum toxin package inserts that neurotoxin immunogenicity is complex. Notably, from the incobotA package insert, incobotA is not immune-inert.
As stated above, we acknowledge the hypothetical nature of some of our discussion points and have now separated basic immunology from BoNT/A-specific aspects. We agree that the detailed mechanisms of formation of neutralizing antibodies to commercial BoNT/A products have not been established yet. However, we think that it is reasonable to logically draw parallels to what is known in immunology and our immune response to bacterial proteins e.g., during infections and vaccinations, and apply this knowledge to BoNT/A products. In particular, the analogy to vaccinations appears to be helpful in illustrating the activation mechanisms of the immune system that are applicable to pharma proteins (including BoNT/A), as well as simplifying and providing context to these complex immunological concepts.
The fact that a neurotoxin is purified to a commercial grade and that only minute amounts of bioactive neurotoxin are injected, does not necessarily mean that such a product cannot activate the immune system. This is nicely demonstrated when one compares the “old” Botox® formulation to the “new” Botox® formulation. Both had been approved by the FDA, yet the old Botox® formulation gave rise to a significantly higher number of immunoresistant cases compared to the new formulation. We’re of the view that the comparison between CPC-BoNT/A products to CPF-BoNT/A products similarly reflects, in part, this “historical” situation. Reducing the load of bacterial proteins – including complexing proteins and flagellin or bacterial DNA, the latter two must be considered as true contaminants – reduces the risk of activating dendritic cells, a necessary first step for the generation of antibodies. We also agree that the doses play a very important role in activating the immune system. However, we would like to mention here that human dendritic cells, due to their function as sentinel cells, are likely the most sensitive responders to minute amounts of microbial components. This holds true especially when these microbial molecules are known ligands for pattern recognition receptors e.g., flagellin for TLR5 and bacterial DNA for TLR9. In that sense, even minute amounts of bacterial components can serve as critical immunologic adjuvants.
We agree that the immune reaction to bacterial pharma proteins is complex, as stated by the FDA. That does not abrogate our objective of trying to explain this mechanism. While the approved on-label aesthetic indications (upper facial lines) for BoNT/A suggest that only minute quantities of purified commercial-grade neurotoxin are being injected into patients, off-label aesthetic use of BoNT/A has expanded to include masseter hypertrophy and body contouring (trapezius, calves, arms), which require significantly higher doses compared to conventional on-label use. As such, the total doses received for aesthetic procedures could easily reach the range used for therapeutic indications. Based on published data from neurology, the frequency of nAb formation ranges from 0.3 - 27.6%, with a higher rate reported for indications requiring higher dose e.g., cervical dystonia and spasticity (Ho WWS, et al. Emerging Trends in Botulinum Neurotoxin A Resistance: An International Multidisciplinary Review and Consensus. Plast Reconstr Surg Glob Open. 2022 Jun 20;10(6):e4407.). Since the immune system does not discriminate between aesthetic and therapeutic use of BoNT/A, one can infer that the rate of nAb formation from aesthetic use will be similar if the total doses administered are comparable.
We also acknowledge that it is true that neutralizing antibodies can develop to CPF-INCO per the package insert. However, one has to consider that the reported cases of nAb formation related to CPF-INCO had all receive a CPC-BoNT/A product (at that time ONA or ABO) previously. In persons who had received CPC-BoNT/A products, switching to CPF-INCO can induce nAb-formation with a low frequency [Dressler D et al. IncobotulinumtoxinA (Xeomin®) can produce antibody-induced therapy failure in a patient pretreated with abobotulinumtoxinA (Dysport(®)). J Neural Transm (Vienna). 2014 Jul;121(7):769-71] (reference 40 in revised manuscriot). However, if one were to refer to several studies cited in our manuscript [28-31], there has been no reported case of antibody-mediated secondary treatment failure in BoNT/A-naïve patients who were treated exclusively from the start with CPF-INCO.
4) The authors should include a Limitations section which is more extensive, and highlights the hypothetical aspects of their proposal. This approach would provide scientific balance to their theories. Among the various limitations, the authors should note that notwithstanding their hypothesis, incobotA is susceptible to antibody formation (see incobotA package insert).
We thank the reviewer for this valuable suggestion. We’ve addressed and discussed in detail the key limitations of our article and proposed approach under the Discussion section. We’ve also suggested possible solutions to try to overcome these limitations in the near future.
We also acknowledge the comment on that it is possible to develop neutralizing antibodies from CPF-INCO use, which the reviewer has rightly referenced to data provided in the FDA product description and package insert. This is true, however, one has to consider that the reported cases of nAb formation related to CPF-INCO had all receive a CPC-BoNT/A product (at that time ONA or ABO) previously. When these studies were performed almost two decades ago, it was not clear whether it would make a difference if naïve (i.e., untreated) patients, or those who had previously received a CPC-BoNT/A product were injected with INCO. In persons who had received CPC-BoNT/A products, switching to CPF-INCO can induce nAb-formation with a low frequency [Dressler D et al. IncobotulinumtoxinA (Xeomin®) can produce antibody-induced therapy failure in a patient pretreated with abobotulinumtoxinA (Dysport(®)). J Neural Transm (Vienna). 2014 Jul;121(7):769-71] (reference 40 in revised manuscriot). However, if one were to refer to several studies cited in our manuscript [28-31], there has been no reported case of antibody-mediated secondary treatment failure in BoNT/A-naïve patients who were treated exclusively from the start with CPF-INCO. To clearly differentiate between the “naïve” situation and “switching” scenario, we’ve made it a point in this manuscript to always state “exclusive use of CPF-INCO” when discussing the risk of nAb formation in relation to INCO.
5) Misuse of nomenclature: The authors inaccurately describe the 150 kDa “core neurotoxin” is the “active [pharmaceutical] ingredient”. The FDA has adopted the USAN-designated non-proprietary name to the Active Pharmaceutical Ingredient (see Karet GB (2019) How Do Drugs Get Named? AMA.J.Ethics 21:E686-E696. NLM: PM:31397664). Consequently, the full complexed toxin is the API.n
We thank the reviewer for pointing this out and have corrected the term “active ingredient” to “active molecule”. We did not intend to misuse the nomenclature and apologize for this. The message that we wanted to convey is that the 150 kDa core toxin (heavy and light chain covalently linked by the disulfide bridge) is necessary and sufficient to inhibit release of neurotransmitter. Therefore, it is the active molecule, alas not pharmacologically defined ingredient. The fact that there are now four approved and commercially available BoNT/A formulations (IncobotulinumtoxinA. DaxibotulinumtoxinA-lanm, Coretox®, and most recently RelabotulinumtoxinA) that contain solely the 150 kDa core neurotoxin as the active molecule demonstrates this. Furthermore, bacterial proteins such as complexing proteins including NTNHA, may be included in CPC-BoNT/A formulations, but are not required for neurotoxicity. To avoid unnecessarily adding on to this manuscript’s length and digressing from the key issues of this manuscript, we did not elaborate on a possible function of complexing proteins either in the food poison or in pharmaceuticals.
6) The authors claim that, in the setting of a protein complex, the associated proteins are “adjuvants.” This contradicts the current guidelines for describing protein complexes (see https://www.ncbi.nlm.nih.gov/genbank/internatprot_nomenguide/; Last updated: 02-MAR-2020). Use the “adjuvant” terminology to describe complexing proteins is not established. Notably, it is highly unlikely that regulatory agencies would approve “a BoNT/A product that contains adjuvants” (line 232). Throughout the article, the authors utilize the word liberally.
We thank the reviewer for drawing our attention to the fact that the term “adjuvant” can have two different meanings, a pharmacological and an immunological one. Our intended definition for the term “adjuvant” in the context of this manuscript refers to immunologic adjuvants, per the Wikipedia citation (Reference 53 in revised manuscript). As such, we’ve replaced the term “adjuvant” with “immunologic adjuvant” throughout the text for clarity. We hope that this is acceptable for the reviewer.
Indeed, an adjuvant from a pharmacological point of view would describe a substance that enforces the genuine pharmacological effect of the active substance, which would be wrong in the immunological context, because immunologic adjuvants are not at all involved in modulating the pharmacological effect (the neurotoxicity of BoNT/A in this case). They do not work on nerve cells, but enhance the immune response at critical activation points (DCs, naïve and memory B cells) as elaborated in this manuscript. In that sense, it is imperative to make the difference clear here. We were not aware of the possibility of misusing the phrase in that respect and apologize for that.
With respect to the liberal use of the term “adjuvant”, we would like to emphasize that “immunologic adjuvants” are at the center of our hypothesis, and this term will be mentioned throughout the manuscript text for reasons aforementioned.
The reviewer also raises a valid point that it is highly unlikely that regulatory authorities would approve a BoNT/A product that contains adjuvants. We agree that regulatory authorities strive at avoiding this. However, the fact is that this happened in the past with the old Botox® formulation; and regulatory authorities e.g., in Korea explicitly warn that some components of BoNT/A products may be immune-stimulatory with the risk of antibody formation, which basically is the function of an immunologic adjuvant. Regulatory authorities make decisions based on best available knowledge of the time at which they have to approve a product. This knowledge may change with additional clinical data becoming available from the long term and daily use of a pharmaceutical. This is one reason why adverse effects are closely monitored. Ultimately, it all comes down to the question of how many cases of immunoresistance are acceptable for a drug at the time of approval. And when it will become necessary to warn of potential new problems also with approved drugs. There are examples of drugs that cause the formation of nAbs in nearly 100% of all patients, still they had been approved by regulatory authorities, one prominent example is Muromonab CD3. Another example was a recombinant Erythropoietin product that caused fatal immune responses in patients due to a new formulation [Schellekens H et al. Erythropoietin-Associated PRCA: Still an Unsolved Mystery. J Immunotoxicol. 2006 Sep 1;3(3):123-30; Boven K et al. Epoetin-associated pure red cell aplasia in patients with chronic kidney disease: solving the mystery. Nephrol Dial Transplant. 2005 May;20 Suppl 3:iii33-40] in which one or more new components functioned as potent immunologic adjuvants causing formation of nAbs to Erythropoietin with fatal outcomes in some patients.
7) The authors further classify the complexing proteins as “unnecessary”. This assertion is not established. They suggest that the complexing proteins are not “bioactive”. Challenges to this approach were recently reviewed by Avelar R (2024) Botulinum Toxin Accessory Proteins: Are They Just an Accessory? Dermatol.Surg. NLM: PM:38864825.
We thank the reviewer for giving us the opportunity to clarify this point.
Although there are publications suggesting that complexing proteins service a function in the process of neuromodulation, such as that mentioned by the reviewer [Avelar R (2024 Dermatol.Surg], the molecular mechanisms remain elusive as to how these associated proteins can influence the interaction of the heavy chain with the SV2 receptor on the surface of the nerve terminal; especially if one accepts that NTNHA is a pH sensitive linker of the core 150kDa BoNT/A to the hemagglutinins [Matsui T et al. Structural basis of the pH-dependent assembly of a botulinum neurotoxin complex. J Mol Biol. 2014 Nov 11;426(22):3773-3782]; and that complexing proteins have been reported to dissociate out of the 900 kDa large complex in a pH dependent manner from the core 150 kDa BoNT/A molecule [Eisele KH et al. Studies on the dissociation of botulinum neurotoxin type A complexes. Toxicon. 2011 Mar 15;57(4):555-65].
The dispensable role of complexing proteins in the process for neuromodulation to occur is also demonstrated by the fact that the four commercially available complexing protein-free BoNT/A formulations function as neuromodulators to produce the intended clinical effect in practice. This topic has been discussed extensively in the past, and it is now widely accepted that complexing proteins are attached to core BoNT/A via NTNHA in a pH- sensitive non-covalent binding. In neutral or weakly basic pH settings, the core BoNT/A will dissociate from the complexing proteins. Therefore, we are of the opinion that claims on complexing proteins affecting the neurotoxic effect of BoNT/A would have to be addressed very critically with present day knowledge.
That being said, the authors do not want to give the impression that complexing proteins do not serve any biological function, as they are definitely involved in the translocation of BoNT/A (= 900 kDa large complex) as a “food poison” through the intestinal barrier [Lee K et al. Molecular basis for disruption of E-cadherin adhesion by botulinum neurotoxin A complex. Science. 2014 Jun 20;344(6190):1405-10]. However, this mechanism of action by complexing proteins is irrelevant in the clinically setting where BoNT/A is injected directly into target tissues.
If one postulates that complexing proteins and BoNT/A might stay together in a physiological tissue pH of around 7 to 7.4 by some unknown mechanism, there could be a modulatory effect e.g., by affecting the speed and area of spread due to the increase in molecular size of the BoNT/A complex. This is one mechanism suggested in the paper referenced by the reviewer, but this activity would presumably not interfere directly with binding and uptake of the core 150 kDa molecule itself. Otherwise, CPC-BoNT/A products would not work or would work less efficiently as CPF-BoNT/A products. It was also proposed that addition of complexing proteins to BoNT/A enhances the proteolytical activity (of the light chain). This is true in an in vitro assay. To our knowledge, the addition of human serum albumin can also result in an increase in proteolytic activity in such an in vitro assay system. However, it is very difficult to comprehend and explain how these in vitro results translate functionally at the tissue level if one accepts the established molecular mechanisms of uptake and translocation of the light chain of BoNT/A (which harbors the protease activity of cleaving SNAP-25) within the nerve cell. How would a complexing protein reach the cytoplasm of the nerve terminal even if it would be taken up together with the core 150 kDa BoNT/A molecule into a recycling vesicle?
However, as this topic is not the focus of our discussion in this manuscript, we have decided to omit the term “unnecessary” to avoid unnecessary distractions to the reader.
8) Line 75: Use of the word “many” is inappropriate as the references included total to 4 with limited primary data (non-review, peer-reviewed publications); “several” would be a more appropriate adjective.
We thank the reviewer for highlighting this point. We exchanged the word “many” for “several” as suggested. The former line 75 is now line 101 in the revised manuscript.
9) Lines 138-141: Suggest refrain from using condescending language.
We believe the reviewer is referring to the following text lines:
“1.4. Breaking immunoresistance by continuing treatment with the same antigen ostensibly contradicts common understanding of the immune system.
These findings seem to contradict the layperson’s understanding of how the immune system, or more precisely, the immunological memory works.”
We apologize for the oversight and in giving such an impression. We certainly did not intend to come across as condescending. As such, we’ve reorganized the manuscript text and Section 1.4. now describes the aims of the article and reads (currently lines 158 – 163):
“1.4. Aims of article
Common notion is that repeated injection with the same antigen, also called “boosting or boostering”, should rather lead to an increase and not to a decrease of antigen-specific Ab titers, a process exploited to improve protection by repeated vaccinations with the antigen. Thus, these findings seem to contradict the layperson’s understanding of how the immune system, or more precisely, the immunological memory works.”
We hope that the reviewer now finds the language neutral and accepts this.
10) Lines 161- 170: Include a reference for these statements.
We thank the reviewer for this comment. The contents in section “2.1. How the immune system becomes activated to produce antibodies” describes basic immunology and the strictly hierarchical manner in which the immune system is activated. Where appropriate we’ve added references from reviews dealing with important aspects of the naïve immune response (currently lines 196 – 198).
In addition, we’ve removed the following sentences to make room for additional explanations (requested by reviewer No.2) and include more citations with respect to the pivotal role of dendritic cells as the gatekeepers of the immune response (currently lines 198 – 205):
“For this to happen, a fixed set of requirements have to be fulfilled: 1) protein antigens need to be processed to peptides, which are presented to naïve T lymphocytes (T cells) by professional antigen presenting cells (APCs); 2) antigen presentation must take place in the special environment of a secondary lymphoid organ such as a lymph node (or spleen); 3) subsequently, antigen-specific T helper lymphocytes must become activated to help anti-gen-specific B lymphocytes (B cells) produce and release antigen-specific Abs.”
We hope that this is acceptable for the reviewer.
11) Figure 1: This figure is very confusing and difficult to follow; several details in this figure are hypothetical. Those proven should be referenced. For example:
- Lines 216-217: Please provide reference of data showing BoNT/A is phagocytosed by DCs
- Lines 218-225: Need references to show that MHC class II presents BoNT/A antigen and that there is clonal expansion (Th and B cell proliferation) is response to BoNT/A exposure.
- Line 225: Has B cell phagocytosis of BoNT/A been demonstrated? Please provide reference.
- Line 230: Please also provide reference that adjuvants amplify clonal expansion
- Line 235: Need reference
- Lines 237-238: Need reference
- Line 255: what are the conditions needed to generate 10E3+ clonal offspring, and has that been demonstrated in routine patients receiving BoNT therapy?
- Lines 293ff: difficult to reconcile with the figure
Thank you for drawing our attention to this. Indeed, Figure 1 can be confusing to the reader as we had combined basic immunology knowledge with known or postulated BoNT/A responses.
We’ve redesigned Figure 1 completely to only depict established basic immunology knowledge without any reference to BoNT/A. We hope that the new Figure 1 is now easier to read.
As mentioned above, we added a new section “2.6. Response to BoNT/A injections in the naïve situation” (currently lines 441 – 546) that addresses what is known and our postulations with respect to BoNT/A-specific aspects of the immune response in the naïve situation. In this new section, we’ve added several new citations to underscore the individual aspects of BoNT/A-specific immune responses. We hope that these new BoNT/A-specific content answers the individual requests for references in relation to the former Figure 1.
12) Figure 2 is mentioned in the body of the article as a general immunological concept but referred to as BoNT/A- specific in the legend of the figure. If this is BoNT/A specific, references are needed to support each statement.
Similar to Figure 1, we have completely redesigned Figure 2 to show only established basic immunology knowledge without any reference to BoNT/A. We hope that the new Figure 2 is now easier to read.
The BoNT/A-specific aspects of the recall situation (immunological memory) are now addressed separately in a new section “3.5. Immunological memory and repeated injections of BoNT/A – role of immunologic adjuvants” (currently lines 604 – 622), where we clearly indicate that the hypothetical nature of this discussion as there is currently no published references to support the BoNT/A-specific statements.
13) Line 454: this statement should include the concept that memory B cells can be reactivated only above a certain threshold and within a certain timeframe, both of which vary depending on the amount and nature of the initial exposure.
We thank the reviewer for giving us the opportunity to clarify this important aspect of reactivating memory B cells. The knowledge that strength of the initial activation signals responsible for the (optimal) activation of dendritic cells, in fact, dictates how strong and how long-lasting B cell response and memory will be, is another fascinating aspect. However, this adds another level of complexity to an already complicated series of events. Therefore, we had originally decided to omit this aspect here for simplicity. Nevertheless, we agree this aspect underscores the pivotal role of DCs and especially immunologic adjuvants, and therefore should be mentioned in this manuscript. We did this by including two additional sentences at the end of section 2.5.3 (currently lines 416 – 420) and citing several key review articles addressing this issue [new references 73-78]:
“It should be mentioned that the reactivation of memory B cells is regulated at several different levels. With respect to immunologic adjuvants, it is important to note that the extent of the initial activation of DCs dictates not only the whole activation process in the naïve situation but also controls the strength of antibody production and longevity of memory B cells [reviewed in 73-78].”
We hope that the reviewer agrees with our proposed revisions to balancing the dilemma of being concise while sufficiently comprehensive.
14) Lines 463, 478 & 542: the authors discuss “dangerous microbe”, “dangerous challenge” and “danger signals or adjuvants” and make the association to BoNT/A. Rather than comparing a licensed pharmaceutical to a “dangerous” substance, the more relevant and less inflammatory wording might be “foreign protein”.
We thank the reviewer for giving us the opportunity to clarify this point.
The “danger theory” or hypothesis was introduced to Immunology by the pivotal publications of Polly Matzinger [1. Matzinger P. Tolerance, danger, and the extended family. Annu Rev Immunol. 1994;12:991-1045; and 2. Matzinger P. The danger model: a renewed sense of self. Science. 2002 Apr 12;296(5566):301-5). This “danger theory” has found its way into Immunology textbooks and it widely accepted [reviewed in Pradeu T, Cooper EL. The danger theory: 20 years later. Front Immunol. 2012 Sep 17;3:287]. The “danger theory” introduced the -at that time- novel aspect that our immune system does not primarily decide to react to something “foreign” but rather to a “danger signal” first. Matzinger proposed that it is not primarily important to distinguish between “own” or “foreign”, but it is much more relevant to be able to identify which type of challenge is “dangerous” for us. And she identified microbes as the most dangerous challenge. Thus, she argued that the immune system first senses danger signals, most prominently microbial surface structures or microbial nucleic acids to respond immediately by a fast reaction of the innate immune system before the delayed adaptive immune system sets in.
In simpler terms, the immune system employs two different criteria to decide whether to mount an immune response to a challenge or not. These criteria are “dangerous” and “foreign”. Two different types of leukocytes function as the decision makers in a strictly hierarchical fashion. The first decision maker is the dendritic cell that is perfectly equipped with pattern recognition receptors (PRRs) that recognize prototypical microbial surface structure or nucleic acids (but also other structures). The sentinel DCs are optimally activated of the in the periphery (skin, mucosa etc.) by engagement of PRRs. This stimulates DCs to phagocytose what they have recognized as being “dangerous”. They digest the proteins to generate peptides. Some of these peptides can be loaded onto MHC molecules and presented on the surface to the second decision maker – naïve T helper lymphocyte in the case of MHC class II presentation. If this T cell recognizes the presented peptide as “foreign”, it will respond in the manner described in the manuscript.
We tried to apply the basics of the “danger theory” to the situation of injecting a mixture of an antigen (BoNT/A) with other bacterial components (these include complexing proteins, flagellin, bacterial DNA, as well as inactive/denatured proteins – all of which have been described to be contained in some CPC-BoNT/A products) into a human being. We reason that it is logical to believe that the immune response to a BoNT/A product should not be in any way different to all other challenges that are introduced to our body. If one follows our line of reasoning, it becomes clear that we’re not suggesting that a licensed pharmaceutical is dangerous per se in a clinical sense; instead, we’re proposing that some of the components in such pharmaceuticals could be recognized as a “danger signal” to stimulate an immune response eventually leading to antibody production. As such, we hypothesize that the lack of such immune-stimulatory bacterial components in CPF-BoNT/A products explains why such products are less immunogenic than those that contain immunological adjuvants.
We fully agree that pure bioactive CPF-BoNT/A is a “foreign” substance as it is of bacterial origin. However, it is not “dangerous” from an immunological point of view as it cannot be recognized by DCs in its pure form (i.e., 150kDa core neurotoxin). The explanation for this is because BoNT/A is an exotoxin, which means it is not expressed on the surface of bacteria, and is not a prototypical danger signal. It does not bind to PRRs nor activate DCs on its own. Therefore, it is not capable of activating the naïve immune system by itself even though it is indeed foreign (it can provide the second signal). In order immune activation to happen, stimuli for PRRs are required that provide the first signal i.e., the “danger signal”. We put forth in our hypothesis that this is the key difference between CPC- and CPF- BoNT/A products. It has not been directly proven, but the real-world data and clinical experience strongly support this hypothesis. The correlation between presence of complexing proteins and immunoresistance has been demonstrated and published [e.g. Hefter H et al. Significantly lower antigenicity of incobotulinumtoxin than abo- or onabotulinumtoxin. J Neurol. 2023 Feb;270(2):788-796.]
As such, we believe that choosing the word “foreign protein” might be misleading and oversimplification of the underlying immunological mechanisms leading to nAb formation. We hope that the reviewer is agreeable with our detailed explanation above.
15) Line 516: Please provide data to support the assertion that CPC products stimulate immunological memory more effectively than CPF-INCO?
We thank the reviewer for giving us the opportunity to try to clarify this point. As mentioned above, this is a working hypothesis that we’ve put forth in this manuscript based on well-established immunological knowledge, as well as clinical data and experience from vaccination. We acknowledge that there is no published data to conclusively support our theories on immune memory specific to BoNT/A and have recognized this as a limitation of this manuscript.
16) Case report: The authors should note that incobotA units are not the same as onabotA Units. The FDA label and additional published data have demonstrated that the Units of onabotA can not be converted into units of incobotA (see Xeomin package insert: “WARNING: Dosing Units of botulinum toxins are not interchangeable between commercial products.” This is supported peer-reviewed literature.
We thank the reviewer for this comment and highlighting the non-interchangeability of botulinum toxin units. We’re aware of this issue that has been discussed extensively in medical literature and practice. One has the impression that this discussion is mainly fuelled by different manufactures rather than by clinicians. Many physicians have agreed that the dose conversion ratio for ONA units to INCO units in clinical practice is 1:1 or close to 1:1. [e.g., Prager W, et al. Botulinum toxin type A treatment to the upper face: retrospective analysis of daily practice. Clin Cosmet Investig Dermatol. 2012;5:53-8], while others see this differently [e.g., Rupp D et al. OnabotulinumtoxinA Displays Greater Biological Activity Compared to IncobotulinumtoxinA, Demonstrating Non-Interchangeability in Both In Vitro and In Vivo Assays. Toxins (Basel). 2020 Jun 13;12(6):393].
The equivalence of INCO and ONA have also been demonstrated in many head-to-head clinical trials utilizing a 1:1 dose conversion ratio [e.g., 1. Sattler G, et. al. Non-inferiority of incobotulinumtoxinA, free from complexing proteins, compared with another botulinum toxin type A in the treatment of glabellar frown lines. Dermatol Surg. 2010;36(Suppl 4):2146-2154; 2. Kane MA, et al. A randomized, double-blind trial to investigate the equivalance of incobotulinumtoxinA and onabotulinumtoxinA for glabellar frown lines. Dermatol Surg. 2015;41(11):1310-1319.]
17) Notably, the patient was exposed to ~40U onabotA per masseter per session (total 80U). The patient was subsequently treated with a total of 50U incobotA. The treatment wore off prior to expected, and the authors concluded that the patient was a secondary non-responder. Notwithstanding non-interchangeablity, this conclusion is not clinically supported since a lower dose of incobotA would be expected to result in a less robust response compared to the administered dose of onabotA. Indeed, when a higher dose of 73U incobotA was administered, the expected response was achieved.
We thank the reviewer for highlighting the gaps and inconsistencies in this case report. We fully agree that the initial working diagnosis of partial SNR to BoNT/A is not clinically supported due to the discrepancies in doses utilized. As the BoNT/A doses received by the patient prior to serological confirmation of nAbs (in August 2019) was sketchy and based purely on recall, we’re unable to get a full and accurate picture of her BoNT/A treatment history until she was followed up regularly at the author’s clinic. In view of the inconsistent clinical history and unclear doses of BoNT/A injected, we’ve revised the case report to only reflect the clinical picture and nAb trend from 2019 onwards (currently lines 633 – 649).
We also like to clarify that the focus of this case report is on the nAb trend while the patient was receiving continuous BoNT/A treatment, so as to demonstrate the low immunogenicity of CPF-INCO and absence of boostering effect even in a patient with existing nAbs. We hope that our explanation and edits are acceptable to the reviewer.
18) Figure 3: The X axis legend is not interpretable. The graph should be annotated with dates of treatment and dose administered.
We thank the reviewer for highlighting this mistake and apologize for the illegible X-axis. We’ve reformatted the graph in Figure 3 by using roman numerals for months and arabic numerals for years e.g. I/23 for January 2023. This allows spreading of the x-axis and should improve its readability.
The dates of treatment coincide with the date of nAb determination and a fixed dose of 50 units per masseter was administered at each visit as mentioned in lines 645 – 649:
“Following this, she continued to receive Xeomin® every 3-4 months, from July 2019 – April 2024, with a dose of 50 units per masseter (total 100 units). At every visit, photographs were taken to monitor clinical response, her serum was obtained and sent for MHDA, fol-lowed by injection of Xeomin® into both masseters. The main objective was to monitor her trend of nAb titers over time with continuous Xeomin® injections.”
We have chosen to describe these in text and omit the details from Figure 3 to prevent it from appearing too cluttered. We hope that this is acceptable to the reviewer.
18) Figure 4: it would be helpful to annotate the figures with the dates as opposed to listing in the legend. Are the figures at maximum contraction? How long after each injection were the figures taken?
We thank the reviewer for the comments to improve Figure 4. We’ve annotated the images as suggested, and will like to clarify that the clinical photos were taken at maximum contraction.
We’ve also include the time points at which the images were taken in the figure legend (currently lines 672 – 678) as follows:
“Figure 4. Assessment of clinical responses of patient. (A) Photographic demonstration of masseter reduction. Before Xeomin® injection in September 2023 and March 2024; 2 months post-injection with Xeomin® in May 2024. (B) Ultrasonographic measurement of masseter thickness reduction. Following Xeomin® injection in January 2024, mean thickness of masseter decreased from 15.7mm to 15.0mm in March 2024, and from 14.7mm in April 2024 to 13.5mm in May 2024. (C) Photographic demonstration of clinical response to Xeomin® treatment in the glabella (left) before; September 2023 (right) after; November 2023).”
We hope that these revisions are acceptable to the reviewer.
19) What was the clinical indication to continue to treat the patient when they were non-responsive to incobotA.
We thank the reviewer for the opportunity to clarify this point. The main purpose of continuing BoNT/A treatment at regular intervals despite the patient being completely non-responsive was to document her nAb titers and assess if INCO provides a “booster” effect. The patient was also keen to continue receiving INCO injections despite the absence of clinical effect, with the hope that some degree clinical effect will return as soon as her nAb titres decline to a level that was sufficiently low. This decision was by the patient after considering all options offered to her by the treating physician, including the cessation of all BoNT/A injections for an extended period of time (i.e., treatment holiday).
20) The authors state that incobotA “does not contribute to” the formation of antibodies. However, the incobotA Package Insert has examples of nAbs. Please explain this contradiction.
We thank the reviewer for this comment, and would like to refer the reviewer to our responses under points 3 and 4 (see above). As mentioned above, the small percentage of cases with nAbs reported in the Xeomin® package insert had also received CPC-BoNT/A products in their treatment history. If patients were treated exclusively with INCO from the start, there has been reported case of nAb-related SNR. The difference lies in naïve versus pretreated patients. In order to clearly differentiate between the “naïve” situation and “switching” scenario, we’ve made it a point in this manuscript to always state “exclusive use of CPF-INCO” when discussing the risk of nAb formation in relation to INCO.
We hope that this explanation is acceptable to the reviewer.
21) Figure 5: This image is hypothetical and should be indicated as such. If this graph represents data from conducted studies, please provide the summarized data points in the article or provide the reference.
I do not think the author’s theory summarized in the figure explains the nAb titer response for one of the three groups of patients where there was an observation of a transient increase in nAb titer following switch to CPF-INCO in the first two years before falling back down again (see text on page 3/27 of the manuscript). In this group of patients, they describe a booster phenomenon with the identical antigen but in the other two groups, they hypothesize the importance of adjuvants and its dual signaling on B-cells. Furthermore, if it acts as a booster, then why would the nAb titers fall after two years? This would be a limitation of the hypothesis.
Also, it was unclear why the relative clinical response was higher on the right vertical axis, but NAbs were higher on the left vertical axis. Graphically, I would anticipate the higher response on the right pointing down, consistent with the lower nAb titre
We thank the reviewer for all the feedback on the original Figure 5. Due to the significant revisions to the manuscript and to incorporate the additional information requested by reviewers, we’ve decided to omit Figure 5 completely. We also acknowledge that the information presented in the original Figure 5 was hypothetical in nature, and its inclusion is not critical to the discussion. We hope that this is acceptable to the reviewer and improves readability of this manuscript.
22) Line 713-714“…since the active neurotoxin is the same in all commercially available formulations…” see comments above regarding nomenclature. In addition, the tertiary structure and post-translatable changes to these complex proteins may not be identical, notwithstanding that the amino acid sequence may be the same.
We agree that differences in tertiary structure might exist, especially as many posttranslational modifications of proteins in bacteria have been identified. [e.g. Macek B et al. Protein post-translational modifications in bacteria. Nat Rev Microbiol. 2019 Nov;17(11):651-664], although to our knowledge this has not been studied in for BoNT/A. To avoid any controversy and confusion, this text passage is no longer mentioned in the new Discussion section of the manuscript.
23) Line 721: Memory B cells and adjuvants were not specifically studied in these references.
We thank the reviewer for highlighting this and agree with the comment. Hence, we’ve removed the references and rephrased the sentence (currently lines 726 – 729) to:
“However, if treatment is restarted with CPF-INCO [35,36], we propose that the activation signals will not be strong enough to re-activate dormant memory B cells due to the absence of immunologic adjuvants.”
We also clearly explain why the cited observations were made, and that this statement is our hypothetical proposal based on what have been discussed in this review. We hope that this is acceptable to the reviewer.
24) Line 723: Please either provide reference or state that this is the author’s hypothetical assessment regarding the threshold of number of molecules of BoNTA. Is the concept based on molar concentration of the total injected molecules?
We acknowledge the reviewer’s comment and have removed this hypothetical assessment in the new Discussion section.
25) Line 726: In the section of limitations, the authors need to address under what circumstances to utilize their approach of continued treatment, and, what potential unintended medical consequences may occur. Measurements of neutralizing antibody titres are not generally available, and therefore the author’s approach is not practical. The approach of continuing incobotA in a patient who is a secondary non-responder can lead to further enhancement of the immune state.
We thank the reviewer for raising these important considerations. We’ve added a completely new part on the limitations of this approach (currently lines 732 – 806) under the Discussion section.

Reviewer 2 Report
Comments and Suggestions for Authors
Dear Authors,
The manuscript is interesting and well presented. It is a significant piece of work regarding an original and clinically relevant topic.
Please find below comments:
Title:
The title is too long, please reduce it: suggestion to remove "why this sometimes work".
Abstract:
Line 8: Antibodies can also target the accessory proteins, not only the core proteins.
Line 11: immunoresistance "occurs"
Line 12: other serotypes BoNT/B (Neurobloc) are alternative treatment, please mention those alternatives.
Line 30: space between "to" and BoNT/A
Introduction:
Line 50: not "transient" but "prolonged" as their effects last for months. Compared to other treatment options, BoNT/A is very persistent.
Line 60: patients can be resistant for other reasons, see Pirazzini et al. 2018: Jul 17;5(8):971-975. doi: 10.1002/acn3.586.Primary resistance of human patients to botulinum neurotoxins A and B.
Line 63: please mention the possible use of BoNT/B.
Line 80: other studies do show insufficient duration of effect with Incobotulinum A which can be due to other type of resistance (Scott Lucchese et al. Cureus. 2024 Feb; 16(2): e53969.). Please review this statement.
Lines 112 to 122: Please provide the proportions of each patient group in %.
Line 149-151: Another question to formulate would be to find out against which antigen the Ab are produced? It could be against other proteins present in the formulations.
Lines 158-159: the authors should consider that some patients may be subject to autoimmunity due to deficient control mechanisms. This theoretical statement should be adapted to real clinical settings.
Line 178: "lectin"
Section 2.2: this section is too long, please reduce the text as the full activation of the DCs is unlikely to happen under BoNT treatment: no inflammation occurs under the injection points usually.
Lines 224-225: please provide definitions of BCR and PRR.
Line 230: please give brief examples for possible adjuvants in the Figure 1 legend and, more specifically in the case of immune reaction against BoNT/A, please provide supportive publications showing the presence of active adjuvants in the BoNT/A formulations in the paragraph below Figure 1 legend (lines 232-243).
Line 250: the release of cytokines is not obvious in the case of BoNT/A injections since no inflammation is seen usually. Please clarify if this applies also to BoNT/A induced immunoreactivity.
Please reduce section 2.3 and 2.4 as the antigen-specific B cell activation do not necessarily apply to BoNT/A immunoresistance unless experimental data already evidence the actions of BoNT/A in this context.
Could the immunological synapse be inhibited by BoNT/A as it would normally block the release of mediators acting on vesicles?
Line 351: contribute "with" IgG antibodies
Line 445 and 449: contribute "with" IgM Abs or IgG Abs
Same comment: the section 3.4 should be reduced as this does not apply completely to BoNT/A immuno-resistance or provide the data in this section to support the hypothesis.
In paragraph between lines 537 and 557: same comment: please do list the potential BoNT/A adjuvants involved or provide supportive data in the paragraph rather than only hypotheses.
Section 4: Case report:
Line 587: it is an "ex vivo" assay not completely in vitro as we do need fresh animal tissues for the assay.
Line 592: space between 50 and units.
Figure 3: Time labels are not clear, please adapt the labels to be easy to read.
Line 600: same, it is an "ex vivo" assay, not in vitro. In vitro is when you do not use any animal but we do need to dissect mouse hemidiaphragm daily for the assay.
Section 4.1-Discussion
Lines 630-631: This point is interesting so please do elaborate on the potential factors, other than Ab response, that may influence the resistance to BoNT/A.
Same comment for Lines 644-645. It would greatly improve the impact of the discussion by elaborating on this topic and quoting other studies.
The discussion is there to position the topic among other studies on the subject.
General comment: please provide more supportive data regarding BoNT/A immunoresistance in the sections 2 and 3. At present, it is only focused on the immune reactions principles and not applied to BoNT/A therapy. This would help the reader to better understand the role of BoNT/A injections.
Author Response
Response to Reviewer 2 Comments
Author's Reply to the Review Report (Reviewer 2)
First, we thank the reviewer spending so much time and effort in carefully reviewing our manuscript, and for the many helpful suggestions to enhance the clarity of the work and make it more useful and appealing to readers. All suggestions and feedback were taken into account and duly considered in the amended version of this manuscript.
As the reviewers have requested for major revisions, we had to reorganize the text and include additional information and references, which resulted in some restructuring of the manuscript. Where possible, we will refer the reviewers to the new position the respective points in the revised version.
Please find our detailed point-by-point responses below, with the reviewer’s comments italicized and bolded for clarity.
Point-by-point Response to Comments and Suggestions for Authors
1) The title is too long, please reduce it: suggestion to remove "why this sometimes work".
We thank the reviewer for this suggestion and have revised the title to “Continuous treatment with IncobotulinumtoxinA despite presence of BoNT/A neutralizing antibodies: Immunological hypothesis and a case report”.
2) Line 8: Antibodies can also target the accessory proteins, not only the core proteins.
We thank the reviewer for the opportunity to clarify this point. It is true that there are antibodies that target the accessory proteins in BoNT/A. However, these antibodies against accessory proteins are not directly involved in blocking the neurotoxic effect of the core BoNT/A molecule. The difference between neutralizing and non-neutralizing antibodies is described in lines 54-56 of the revised manuscript. We hope that this explanation is acceptable for the reviewer.
3) Line 11: immunoresistance "occurs".
We thank the reviewer for highlighting this and have corrected the term. Now line 11.
4) Line 12: other serotypes BoNT/B (Neurobloc) are alternative treatment, please mention those alternatives.
We thank the reviewer for raising the point on BoNT/B. We agree that BoNT/B is an alternative option for patients with immunoresistance, albeit with even higher rates of nAb formation than BoNT/A. We have included this piece of information in lines 78 to 81 of the revised manuscript:
“Switching patients to BoNT/B is a possibility but tends to be only a short-term alternative due its more pronounced immunogenicity compared to BoNT/A, with a high proportion of patients eventually developing nAbs to BoNT/B [13 and reviewed in 3,4].”
We hope that this is acceptable for the reviewer.
5) Line 30: space between "to" and BoNT/A
We thank the reviewer for highlighting this and have corrected the typographical error.
6) Line 50: not "transient" but "prolonged" as their effects last for months. Compared to other treatment options, BoNT/A is very persistent.
We thank the reviewer for for this comment. We agree that the effects of BoNT/A can persist for several months, which is very long for a pharma protein. Nevertheless, the effects are not permanent and repeated injections of BoNT/A are required to maintain its clinical effects. As such, we’ve rephrased the sentence to (currently lines 51 to 53):
“The clinical effects of BoNT/A typically last for months and BoNT/A has to be re-injected at regular time intervals, frequently for many years, if treatment results are to be maintained.”
We hope that this is acceptable for the reviewer.
7) Line 60: patients can be resistant for other reasons, see Pirazzini et al. 2018: Jul 17;5(8):971-975. doi: 10.1002/acn3.586. Primary resistance of human patients to botulinum neurotoxins A and B.
We thank the reviewer for drawing our attention to these important facts. We have addressed this point in lines 59-65 of the revised manuscript:
“Additionally, one must clearly differentiate primary and secondary immunoresistance. Primary immunoresistance is rarely seen in patients [reviewed in 2] [2=Dressler D. Mov Disord. 2004] and the existence of nAbs is commonly ascribed to an exposure of such persons to BoNT/A in the past, e.g. by food poisoning or active immunization (vaccination). These persons do not have any clinical response to BoNT/A injections right from the start. Secondary immunoresistance, in contrast, describes the situation in which a patient had initially responded well to the treatment but the desired effects started to diminish over time.”
We’ve refrained from citing Pirazzini et al. here because this paper deals predominantly with the possibility of mutations being responsible for primary resistance, which is not relevant to the main discussion of this manuscript. We hope that this is acceptable for the reviewer.
8) Line 63: please mention the possible use of BoNT/B.
We thank the reviewer for raising the point on BoNT/B, and have mentioned it in lines 78 to 81 of the revised manuscript:
“Switching patients to BoNT/B is a possibility but tends to be only a short-term alternative due its more pronounced immunogenicity compared to BoNT/A, with a high proportion of patients eventually developing nAbs to BoNT/B [13 and reviewed in 3,4].”
We hope that this is acceptable for the reviewer.
9) Line 80: other studies do show insufficient duration of effect with Incobotulinum A which can be due to other type of resistance (Scott Lucchese et al. Cureus. 2024 Feb; 16(2): e53969.). Please review this statement.
We thank the reviewer for highlighting this point. The aforementioned publication of Luchese et al.is important, but focuses predominantly on the difference in injection-site pain between ONA and INCO in a cohort of 42 patients with chronic migraine. In this article, the only reference with respect to resistance (not immunoresistance!) is a small paragraph at the end of the discussion referring to another publication, in which it was suggested that a more rapid loss of efficacy is observed in a small proportion of patients treated with INCO (termed “insufficient duration of effect”). Even if one wants to suggest that this is indicative of partial resistance, it clearly is not due to nAb formation and cannot be considered “immunoresistance”. We’ve therefore decided not to reference this manuscript. We hope that this is acceptable for the reviewer.
10) Lines 112 to 122: Please provide the proportions of each patient group in %.
We thank the reviewer for this suggestion. We’ve added the percentages of each group in lines 119 – 140) of the revised manuscript.
11) Line 149-151: Another question to formulate would be to find out against which antigen the Ab are produced? It could be against other proteins present in the formulations.
We thank the reviewer for this comment, and agree that this would be another interesting question to address. Studies by Atassi et al (Atassi MZ. Basic immunological aspects of botulinum toxin therapy. Mov Disord. 2004 Mar;19 Suppl.; Atassi MZ. Molecular basis of immunogenicity to botulinum neurotoxins and uses of the defined antigenic regions. Toxicon. 2015 Dec 1;107(Pt A):50-8. doi: 10.1016/j.toxicon.2015.06.003. Epub 2015 Jun 16.) have described the possible antigenic structures on BoNT/A heavy and light chains in great detail. To manage the length of this manuscript, we’ve however, decided to remove this part in the revised manuscript, as the questions are individually addressed in the new section “2. How the immune system is activated – the naïve situation”.
12) Lines 158-159: the authors should consider that some patients may be subject to autoimmunity due to deficient control mechanisms. This theoretical statement should be adapted to real clinical settings.
We thank the reviewer for this suggestion. We’ve decided to remove this sentence (lines 158 – 159 in the original manuscript) to manage the length of the revised manuscript. While the aspects of autoimmunity are interesting, they are not in relevant in the context of nAb formation and BoNT/A immunoresistance. As such, this sentence was omitted to avoid unnecessary distraction to the reader by a completely new (and complex) topic on autoimmunity.
We hope that this is acceptable for the reviewer.
13) Line 178: "lectin”.
We thank the reviewer for highlighting this and have corrected the typographical error. Now line 204.
14) Section 2.2: this section is too long, please reduce the text as the full activation of the DCs is unlikely to happen under BoNT treatment: no inflammation occurs under the injection points usually.
We thank the reviewer for giving us the opportunity to clarify this point.
We understand that Section 2.2. is relatively long and it may seem attractive to reduce the text here. However, we’d like to emphasize that the activation of dendritic cells is the first and most important step in the activation of the whole immune system. Optimal activation of DCs is an absolute prerequisite for all steps that follow, most prominently professional antigen presentation. As immunologic adjuvants are working on this level first, we think we need to stress the pivotal position of DCs in the innate, and later in the transition to the adaptive immune response. We considered removing some aspects, such as the role of DCS in organizing an acute local inflammatory response, but felt that it is better to retain these information to deliver a comprehensive overview of the central role of DCs. For a long time, this central role of DCs had not been recognized, and even today, many physicians are not aware that it is the optimally activated DC that governs all following steps in an immune response, and we think that BoNT/A is no exception. Therefore, we’d like to politely decline with reviewer’s suggestion in shortening section 2.2.
15) Is there inflammation at the site of injection?
We thank the reviewer for giving us the opportunity to try to clarify this point.
Yes, there always is a local short-lived inflammatory response at the site of injection. This does not necessarily produce visible classical signs of inflammation such as redness, swelling, pain or elevated temperature (fever). Still the immune reaction of the local sentinel cells, such as resident macrophages and dendritic cells, is always a stereotypical one that is characterized by the local release of mediators including pro-inflammatory cytokines such as IL-1 and TNF, enzymes and anti-microbial substances. This can be locally and timely restricted, but it does occur. It has been demonstrated that insertion of a sterile needle and injection of a sterile physiological salt solution results in rapid recruitment of first, neutrophilic granulocytes, and then monocytes / macrophages to the injection canal [for a beautiful example refer to: Peters NC et al. In vivo imaging reveals an essential role for neutrophils in leishmaniasis transmitted by sand flies. Science. 2008 Aug 15;321(5891):970-4; and the videos to this publication available at https://www.science.org/doi/10.1126/science.1159194?url_ver=Z39.88-2003&rfr_id=ori:rid:crossref.org&rfr_dat=cr_pub%20%200pubmed]. The recruitment of these leukocytes from the blood stream to the site of injection is the consequence of a local acute inflammatory response that normally does not result in clinical symptoms after (sterile) injection of BoNT/A. Yet, the cells at the injection site, including DCs, are always exposed to this pro-inflammatory milieu, unless one co-injects anti-inflammatory drugs at the same time at the same site.
Tissue destruction by the needle cannot be avoided. Destruction of cells is a non-physiological situation that always results in the release of intracellular contents into the extracellular milieu. Heat shock proteins and other proteins function as endogenous alarm mediators or damage associated molecular patterns that alert and partly activate macrophages and DCs, if the stimulus is strong and long enough. In combination with additional immunologic adjuvants, such as TLR ligands, optimal DC activation may occur especially after injecting a BoNT/A product that contains substances that can activate PRRs.
In conclusion, if one injects BoNT/A (or anything else), she/ he prepares the field for sentinel cell activation and a local timely restricted inflammatory response. We’ve added a short paragraph (currently lines 456- 466) to address this critical point.
We hope that this explanation convinces this reviewer of the importance of DC and DC activation with respect to injection of BoNT/A. We also hope that the reviewer agrees that a subclinical inflammatory process will always result as consequence of injecting BoNT/A (or anything else), although this may not (always) be visible.
16) Lines 224-225: please provide definitions of BCR and PRR.
We thank the reviewer for highlighting the use of abbreviations before properly defining them in the text. We amended this in the revised manuscript and provided definitions for BCR (= B cell antigen receptor) and PRR (= pattern recognition receptor) in lines 248 and 258 of the amended Figure 1.
17) Line 230: please give brief examples for possible adjuvants in the Figure 1 legend and, more specifically in the case of immune reaction against BoNT/A, please provide supportive publications showing the presence of active adjuvants in the BoNT/A formulations in the paragraph below Figure 1 legend (lines 232-243).
We thank the reviewer for this suggestion.
We’ve provided one prominent example of a possible adjuvant (flagellin) for the sake of brevity in the legend to Figure 1 (currently line 248):
“Upon encountering a microbial challenge, microbial surface structures, e.g. flagellin, act as immunologic adjuvants (red arrow) to activate pattern recognition receptors (PRRs) on DCs.”
More information on immunological adjuvants is provided in the new section “2.6. Response to BoNT/A injections in the naïve situation”, which focuses on BoNT/A-specific aspects of innate immune activation. Here, we describe possible adjuvants contained in some BoNT/A products in detail (now lines 442 – 455).
18) Line 250: the release of cytokines is not obvious in the case of BoNT/A injections since no inflammation is seen usually. Please clarify if this applies also to BoNT/A induced immunoreactivity.
We thank the reviewer for this comment. As mentioned in our response to point 15, the local release of cytokines is unavoidable at the site of an injection due to the unavoidable destruction of tissue cells. We kindly refer the reviewer to our detailed explanation above on the acute local inflammatory response to BoNT/A injections.
18) Please reduce section 2.3 and 2.4 as the antigen-specific B cell activation do not necessarily apply to BoNT/A immunoresistance unless experimental data already evidence the actions of BoNT/A in this context.
We thank the reviewer for this suggestion to reduce the length of our manuscript.
Following the recommendations of all peer reviewers, we’ve now separated basic immunological concepts from BoNT/A-specific aspects completely in the new version of the manuscript. Section “2.6. Response to BoNT/A injections in the naïve situation” now deals exclusively with what is known about the respective immune activation steps with respect to BoNT/A. In section 2.6. we provide published data on BoNT/A where available, and clearly state what are based on our hypothetical interpretations. In summary, we hope to convince the reviewer that the aspects discussed in sections 2.3 and 2.4. are also relevant to understand the basic immunological mechanisms that might lead to nAb formation when injecting BoNT/A. As such, we are reluctant to reduce these sections, as they are essential to understand the underlying mechanisms. We hope that this explanation is acceptable to the reviewer.
19) Could the immunological synapse be inhibited by BoNT/A as it would normally block the release of mediators acting on vesicles?
We thank the reviewer for this very interesting question. Frankly, we do not know the answer to this yet. It is conceivable that BoNT/A light chain as a SNPA25 protease would also be able to interfere with the release of other vesicle contents as long as the SNARE complex is involved in this release. However, somehow the light chain would have to get into the cytosol of that respective responder cell (here an immune cell). There are very view reports on (direct?) effects of BoNT/A on immune cells [e.g. Zou YP et al. Botulinum toxin type A inhibits M1 macrophage polarization by deactivation of JAK2/STAT1 and IκB/NFκB pathway and contributes to scar alleviation in aseptic skin wound healing. Biomed Pharmacother. 2024 May;174:116468 and Al-Saleem FH et al. RBC Adherence of Immune Complexes Containing Botulinum Toxin Improves Neutralization and Macrophage Uptake. Toxins (Basel). 2017 May 19;9(5):173]. However, neither receptors for BoNT/A on these immune cells have been characterized nor an uptake demonstrated.
In addition, the molecules involved in forming immunological synapses are usually already present on the plasma membrane of the immune cells facing towards the partner cell type. They are normally not brought to the surface by vesicle fusion like the receptor SV2 of BoNT/A at the nerve terminal. Of course, some of them can be modulated from the inside, such as integrins. However, nothing is known about BoNT/A being involved in these processes.
To avoid lengthening an already long manuscript, we’ve decided to omit this discussion on the immunological synapse in the revised manscript, as it is not mandatory for the understanding of the critical steps leading to nAb production. We hope that the reviewer is agreeable with our decision. Still the question is a very interesting one that warrants further investigation.
20) Line 351: contribute "with" IgG antibodies.
We thank the reviewer for highlighting this and have corrected the typographical error (currently line 335).
21) Line 445 and 449: contribute "with" IgM Abs or IgG Abs.
We thank the reviewer for highlighting the typo. This part has been reorganized to save space.
22) Same comment: the section 3.4 should be reduced as this does not apply completely to BoNT/A immuno-resistance or provide the data in this section to support the hypothesis.
We thank the reviewer for this comment. Kindly refer to our responses above to point 18. We feel that reducing this section would not be helpful in our attempt to explain the basic immunological principles necessary to understand how antibodies are generated in general. We’ve also explained how these basic immunological concepts are applicable to BoNT/A in the new section “3.5. Immunological memory and repeated injections of BoNT/A – role of immunologic adjuvants”.
We hope that this explanation is acceptable to the reviewer.
23) In paragraph between lines 537 and 557: same comment: please do list the potential BoNT/A adjuvants involved or provide supportive data in the paragraph rather than only hypotheses.
We thank the reviewer for the comment, and will like to refer him/her to our response to point 17 above. We’ve added a comprehensive list of possible adjuvants and provided the supporting references in lines 442-455 of the revised manscript:
“2.6.1. Activation of DCs by BoNT/A injections
If a BoNT/A product containing components that may serve as immunologic adju-vants is injected, resident DCs may be activated. Different components in CPC-BoNT/A products can activate the immune system. Flagellin, reported to be contained in Abobotu-linumtoxinA [19,25], is a potent activator of human DCs by binding to TLR5 [reviewed in 80]. Flagellins are being developed as adjuvants in vaccines [reviewed in 81]. Bacterial DNA, reported to be contained in OnabotulinumtoxinA [26] is a potent immune stimulator via TLR 9 [reviewed in 82,83]. Parts of bacterial DNA are employed as adjuvants for vaccines [reviewed in 84]. Clostridial hemagglutinins, also referred to as complexing proteins, have been identified as immune stimulators [85-91]. In addition, some BoNT/A products contain inactive toxin molecules [92]. Inactive proteins are frequently denatured or proteo-lytically cleaved and tend to form protein aggregates. Protein aggregates are a major cause for antibody formation against pharma proteins [reviewed in 93-95] as they activate scav-enger receptors on DCs.”
24) Line 587: it is an "ex vivo" assay not completely in vitro as we do need fresh animal tissues for the assay.
We thank the reviewer for pointing this out and fully agree that the MHDA is not an in vivo assay, is not a complete in vitro assay, and utilizes freshly isolated tissue from sacrificed animals. As such, “ex vivo” is most appropriate. We edited the term accordingly in line 641 of the revised manuscript.
25) Line 592: space between 50 and units.
We thank the reviewer for highlighting this. The typo has been corrected (currently line 646).
26) Figure 3: Time labels are not clear, please adapt the labels to be easy to read.
We thank the reviewer for highlighting this mistake and apologize for the illegible X-axis. We’ve reformatted the graph in Figure 3 by using roman numerals for months and arabic numerals for years e.g. I/23 for January 2023. This allows spreading of the x-axis and should improve its readability.
27) Line 600: same, it is an "ex vivo" assay, not in vitro. In vitro is when you do not use any animal but we do need to dissect mouse hemidiaphragm daily for the assay.
We thank the reviewer again for this comment. We edited “in vitro” to “ex vivo” in line 655 of the revised manuscript.
28) Lines 630-631: This point is interesting so please do elaborate on the potential factors, other than Ab response, that may influence the resistance to BoNT/A.
We thank the reviewer for this comment and the opportunity to elaborate on this point.
First, we want to highlight that the section “4.1. Discussion of case report” (previously lines 623 – 660) in the original manuscript has been shifted under section “4.2. Summary of case report” (currently lines 680 – 696) in the revised manuscript.
There was an apparent period of delay between the resolution of nAb titers and return of clinical response in this patient. After nAb titres have declined to undetectable levels, it is unlikely that the observed lack of clinical resistance is secondary to BoNT/A immunoresistance. Other possible causes of secondary non-response include disease progression, inadequate dosage, incorrect muscle target, or improper injection technique (Ho WWS, et al. Emerging Trends in Botulinum Neurotoxin A Resistance: An International Multidisciplinary Review and Consensus. Plast Reconstr Surg Glob Open. 2022 Jun 20;10(6):e4407). This clinical observation might be partly due to the subjective perception of both patient and physician, and insufficient blockage of all neuromuscular junctions in the masseter muscle as described in lines 691 – 694 of the revised manuscript:
“Clinical response was seen sooner in the glabella (Fig. 4C) from September 2023 onwards compared to the masseters (Fig. 4A). This could be due to the difference in size of the muscles – significantly more neuromuscular junctions need to be blocked in the masseters compared to the glabella for visible results.”
While clinical response in the glabellar is easier to assess clinically, it might not be as apparent in the masseter. Clinical response for the masseters was clearly documented when quantitative and objective measurement of masseter thickness was performed with ultrasonography.
We hope that this explanation is acceptable to the reviewer.
29) Lines 644-645. It would greatly improve the impact of the discussion by elaborating on this topic and quoting other studies.
We thank the reviewer for this comment and the opportunity to elaborate on this point.
First, we want to highlight that lines 644 – 645 in the original manuscript has been shifted under section “5. Discussion” (currently lines 735 – 737) in the revised manuscript:
“Inter-individual differences in immune response exist and many possible reasons have been proposed [reviewed in 130].”
We have elaborated on how genetics might contribute to inter-individual differences through individual MHC expression and ability of antigen presenting cells to present peptides to naïve T lymphocytes, as well as the differential expression and genetic predisposition of pattern recognition receptors (currently lines 740 – 761).
30) General comment: please provide more supportive data regarding BoNT/A immunoresistance in the sections 2 and 3. At present, it is only focused on the immune reactions principles and not applied to BoNT/A therapy. This would help the reader to better understand the role of BoNT/A injections.
We thank the reviewer for this comment and acknowledge the hypothetical nature of some our discussion that combines the basic concepts of immunology and their application to BoNT/A responses.
We have addressed this issue by separating these two aspects completely by introducing new sections “2.6. Response to BoNT/A injections in the naïve situation” and “3.5. Immunological memory and repeated injections of BoNT/A – role of immunologic adjuvants”. We’ve cited more peer-reviewed literature supporting our BoNT/A-related interpretations, and clearly state where our interpretations were hypothetical and derived through the application of basic immunologic principles from other antigens to BoNT/A. We hope this has improved the readability of our manuscript and is acceptable to the reviewer.

Round 2
Reviewer 1 Report
Comments and Suggestions for Authors
Review on Resubmission:
The authors made the major revisions as requested. They acknowledge that much of their manuscript is based on hypothetical extensions of basic immunological theory, the biggest weakness remains that much of their narrative is not supported with evidence when applied specifically to botulinum toxin preparations. One example is the quite categorical statement (see lines 468ff) that the 150kD molecule is "unable to activate dendritic cells by itself" and that therefore "peptides thereof will not be presented to T lymphocytes". The theory might be reasonable, in that effective DC/APC activation is foundational to generation of a strong immune response, but I would ordinarily expect such a conclusion (and the implied benefit to 150kD) to be supported with appropriate evidence that has specifically demonstrated a lack of DC activation following exposure to 150kD BoNT/A vs complexed BoNT/A. Just prior to that for example, they mention that the physical procedure of injection alone results in "alarm signals" that may contribute to activation of dendritic cells, "irrespective of the presence of immune-stimulatory components in BoNT/A products". Who's to say that this level of DC activation isn't sufficient in a clinical setting?
Below are some specific recommendations:
Line 19: Please add the word “potentially” prior to “immune-stimulatory.
Line 32: Please add the word “possible” before the word “alternative”. This is supported by the limitations.
Line 36: Please add “observed in our case described herein” before “is the presence”
Line 41: please replace “will” with “may”
Line 44: unless a non-proprietary name is the first word in a sentence, the word should not be capitalized. For instance, in this line, please use “incobotulinumtoxinA” instead of “IncobotulinumtoxinA”.
Line 61: please add “unknowingly” before “exposure. In line 62, you would want to add a comma after “poisoning”.
Line 88ff: see note above about non-proprietary names.
Line 510: “of with” – a word may be missing, or, “with” does not belong
Line 607: add “potential” prior to “immune stimulatory”
Line 623ff: either in the body of the clinical history or elsewhere, please stipulate non-interchangeability across products, particularly INCO and ONA, as it is listed in product labeling. Otherwise, which readers may find the information confusing. There is sufficient literature to cite that supports non-interchangeability.
Line 637-639: per prior review, and the author’s response, the diagnosis of complete SNR is not supported by the information provided. The authors need to provide some context or opinion on the diagnosis, as the reader may be misled that providing a lower dose of Xeomin would result in a diagnosis of SNR.
Line 644: most readers do not have a context for the meaning of 5 mIU/ml. Please provide some context on this result.
Line 666: the photos are poor quality with insufficient contrast between the background and the patient’s face. Please provide better photos.
Line 673, 677 and 683: please use “incobotulinumtoxinA” instead of Xeomin, here and elsewhere.
Line 677: please stipulate here whether the photos are with maximum frown
Line 680: please provide some context on why the treating physician continued to treat this patient in the presence of immunoresistance.
Line 695: please change “does” to “did not, in this case,”
Line 702: please add “presumably” prior to “because”
Line 705: Please add “although relatively short observation periods” after “[28]”.
Line 734: The author asks why not all patients respond to INCO. The author needs to acknowledge explicitly that that their hypothesis may not be applicable to most patients. Indeed, clinical experience is that patients who are resistant to one BoNTA are resistant to all BoNTAs, including INCO.
Line 776: please reflect that one of the potential issues is related to increased immunogenic response. Consequently, “…this translates to potentially increased stimulation of the immune system, unnecessary…” This is supported by the author’s statement later in the paragraph with the statement “…most likely due to a boostering of memory B cells despite the lack of immunogenic adjuvants.”
Line 794: The author states “The ex vivo MHDA… is very reliable”. Please cite an assessment of the reliability of the MHDA. Are the false positives and negatives with the MHDA known when comparing to clinical outcomes?
Line 797: “detect Abs” – are they referring to neutralizing or binding antibodies?
Line 805: since an ELISA binding antibody assay is not diagnostic, but used as a screening assay, the MHDA is the key confirmatory step. The author should stipulate that the MHDA, or a comparable bioassay (e.g., the mouse neutralization assay), is the “critical” or “key” next step rather than “can be the next step. In the discussion of the MHDA, the author should further stipulate and cite the several studies that have reported that a positive nAB does not necessarily correlate to, nor predict SNR.
Line 823ff: In the author’s summary, the author should acknowledge the other limitations of the practical application of their approach. This includes the potential to further boost the immune response.
With these changes, I believe that the author will have a more balanced publication.
Author Response
Response to Reviewer 1 Comments
Author's Reply to the Review Report (Reviewer 1 – second round)
First, the authors would like to thank the reviewer for having invested again so much time and effort to carefully review our resubmitted manuscript. The suggestions made to improve our manuscript and provide a more balanced viewpoint are highly appreciated.
All suggestions and feedback were taken into account and duly considered in the amended version of this manuscript.
For the sake of clarity and readability, parts that have been amended following the recommendations of the reviewer are highlighted in green.
Please find our detailed point-by-point responses below, with the reviewer’s comments italicized and bolded for clarity.
Point-by-point Response to Comments and Suggestions for Authors
1) The authors made the major revisions as requested. They acknowledge that much of their manuscript is based on hypothetical extensions of basic immunological theory, the biggest weakness remains that much of their narrative is not supported with evidence when applied specifically to botulinum toxin preparations. One example is the quite categorical statement (see lines 468ff) that the 150kD molecule is "unable to activate dendritic cells by itself" and that therefore "peptides thereof will not be presented to T lymphocytes". The theory might be reasonable, in that effective DC/APC activation is foundational to generation of a strong immune response, but I would ordinarily expect such a conclusion (and the implied benefit to 150kD) to be supported with appropriate evidence that has specifically demonstrated a lack of DC activation following exposure to 150kD BoNT/A vs complexed BoNT/A. Just prior to that for example, they mention that the physical procedure of injection alone results in "alarm signals" that may contribute to activation of dendritic cells, "irrespective of the presence of immune-stimulatory components in BoNT/A products". Who's to say that this level of DC activation isn't sufficient in a clinical setting?
We thank the reviewer for this comment. Indeed, parts of our discussion are (still) hypothetical in nature, unfortunately, due to the lack of original data and published evidence specific to botulinum toxin. We tried to take that into account by re-phrasing the critical phrases.
Lines 469-476 now read:
“However, in the absence of immunologic adjuvants, DCs will not be stimulated to perform phagocytosis and a pure bioactive BoNT/A molecule will not be taken up. As an exotoxin that is released by the bacteria, it is not a prototypical ligand for PRRs and is most likely unable to activate DCs by itself. If there is no DC activation peptides thereof will not be presented to T lymphocytes. This is a plausible explanation why injection of a pure and bioactive CPF-BoNT/A product does not induce antibody production, as has been observed in more than a decade of clinical practice in patients that were treated exclusively with CPF-INCO [28-33].”
The reviewer further comments that “the physical procedure of injection alone results in… Who’s to say that this level of DC activation isn't sufficient in a clinical setting?”
Although it can never be excluded completely that in an extremely rare situation, the procedure of injection alone (e.g. with a non-sterile needle) could lead to a full activation of a local DC, it seems rather unlikely that this is the case in real life. Otherwise, one would expect nAb formation with a much higher frequency with all BoNT/A products, irrespective of the absence or presence of immunologic adjuvants. Our intention was to indicate the signal integrator function of DCs (also in part to respond to a request of reviewer 2).
We hope that our explanation is sufficiently clear and the reviewer agrees with our proposed revisions.
2) Line 19: Please add the word “potentially” prior to “immune-stimulatory.
We thank the reviewer for the suggestion and have added “potentially” as recommended (now line 20).
3) Line 32: Please add the word “possible” before the word “alternative”. This is supported by the limitations.
We thank the reviewer for the suggestion and have added “possible” as recommended.
3) Line 36: Please add “observed in our case described herein” before “is the presence”
We thank the reviewer for the suggestion and have added “observed in our case described herein” as recommended.
4) Line 41: please replace “will” with “may”
We thank the reviewer for the suggestion and have replaced “will” with “may” in line 41.
5) Line 44: unless a non-proprietary name is the first word in a sentence, the word should not be capitalized. For instance, in this line, please use “incobotulinumtoxinA” instead of “IncobotulinumtoxinA”.
We thank for the reviewer for highlighting this and have corrected the typographical mistakes. All non-proprietary names (including onabotulinumtoxinA and abobotulinumtoxinA) in the manuscript text are now not capitalized
6) Line 61: please add “unknowingly” before “exposure. In line 62, you would want to add a comma after “poisoning”.
We thank the reviewer for the suggestion and have added “unknowing” before “exposure” (now line 62), as well as a comma after “poisoning” (now line 63).
7) Line 88ff: see note above about non-proprietary names.
We thank for the reviewer for highlighting this and have made the necessary edits throughout the manuscript (see response to comment 5). All non-proprietary names have also been abbreviated henceforth as ONA, ABO, and INCO.
8) Line 510: “of with” – a word may be missing, or, “with” does not belong
We thank for the reviewer for highlighting the typo and have removed “with”.
9) Line 607: add “potential” prior to “immune stimulatory”
We thank the reviewer for the suggestion and have added “potential” as recommended (now line 608).
10) Line 623ff: either in the body of the clinical history or elsewhere, please stipulate non-interchangeability across products, particularly INCO and ONA, as it is listed in product labeling. Otherwise, which readers may find the information confusing. There is sufficient literature to cite that supports non-interchangeability.
We thank the reviewer for this comment and the opportunity to clarify this point.
It was not our intention to imply that the BoNT/A units were interchangeable across product and have added the following in line 653 – 655:
“It should be noted here that in general the units of different BoNT/A products are not interchangeable (see FDA approvals of INCO and ONA [20,21]) [reviewed in 126-128].”
However, we would like to draw to attention that there are several publications showing that the dose conversion ratio of INCO and ONA in daily clinical practice is 1:1 or close to 1:1 (e.g., Dressler D, et al. Measuring the potency labelling of onabotulinumtoxinA (Botox®) and incobotulinumtoxinA (Xeomin®) in an LD50 assay. J Neural Transm (Vienna). 2012;119(1):13–15; Prager W, et al. Comparison of two botulinum toxin type A preparations for treating crow’s feet: a split-face, double-blind, proof-of-concept study. Dermatol Surg.2010;36 Suppl 4:2155–2160; Kane MA, et al. A randomized, double-blind trial to investigate the equivalence of IncobotulinumtoxinA and OnabotulinumtoxinA for glabellar frown lines. Dermatol Surg.2015;41(11):1310–1319). We also acknowledge that there are several other publications that claim different dose conversion ratios, but we refrain from discussing this controversial topic in greater detail as it is not the focus of this manuscript.
11) Line 637-639: per prior review, and the author’s response, the diagnosis of complete SNR is not supported by the information provided. The authors need to provide some context or opinion on the diagnosis, as the reader may be misled that providing a lower dose of Xeomin would result in a diagnosis of SNR.
We thank the reviewer for this comment and agree that the clinical history is insufficient to suggest a diagnosis of complete SNR, especially when a lower dose of INCO injected. We certainly did not mean to mislead the reader and our intention was to illustrate how the clinical suspicion of nAb-induced SNR was reached after 2 failed treatment sessions. We have revised this part of the text to reflect this point more clearly by removing the potentially confusing statement on how the diagnosis of complete SNR was made clinically (now lines 638-644):
“She was injected again with 34 units of INCO to each masseter. The treatment still yielded no clinical results, and she was suspected to have developed immunoresistance to BoNT/A. The patient’s serum was subsequently sent to a specialized laboratory (toxogen GmbH, since January 2023 toxologics GmbH, Hannover, Germany) in August 2019 for an ex-vivo mouse hemidiaphragm assay (MHDA) which confirmed the presence of nAbs to BoNT/A and clinical diagnosis of nAb-induced complete SNR.”
The diagnosis of nAb-induced SNR was only made definitively after MHDA was performed.
We hope that this is acceptable for the reviewer.
12) Line 644: most readers do not have a context for the meaning of 5 mIU/ml. Please provide some context on this result.
We thank the reviewer for highlighting this point. It is indeed not self-explanatory what “5 mIU/ml” mean.
We have inserted a short explanation on the upper and lower detection limits of the MHDA to provide some context on what the absolute values represent (now lines 645 – 647):
“The highly sensitive MHDA’s cut-off point for nAb detection is 1.82 mIU/ml and up to a maximum of 12.15 mIU/ml [4, 36, 41, A.Rummel personal communication)]. The patient’s nAb titre returned as 5 mIU/ml."
We have also modified lines 659 - 665 into:
“The patient’s nAb titers were progressively on a downward trend from July 2019, eventually reaching below the MHDA’s lower cut-off point of 1.82 mIU/ml in December 2022 (Fig. 3) and was hence undetectable.”
We’ve also corrected the date from January 2023 to December 2022 as the time at which the blood was drawn. The MHDA was performed in January 2023.
Finally, we have updated Figure 3. Here, the y-axis of Figure 3 was changed into “BoNT/A neutralizing potency [mIU of reference antisera/ml]. The amended legend provides more information on the MHDA for the reader (now lines 664-673):
Figure 3. Trend of nAb titers over time. From July 2019 (VII/19) to January 2024 (I/20), the patient’s serum was analyzed for nAbs to BoNT/A using an ex-vivo mouse hemidiaphragm assay (service performed by toxogen GmbH, since Jan 2023 by toxologics GmbH, Hannover, Germany) at regular intervals. The patient received INCO every 3-4 months, from July 2019 (VI/19)– April 2024 (IV/24), with a dose of 50 units per masseter. Titers of nAbs were progressively on a downward trend from July 2019 (VI/19), eventually reaching below the lower cut-off point of 1.82 mIU/ml and hence regarded as undetectable. in January 2023. (International unit (IU)/ml is a measurement of neutralizing BoNT/A activity in serum. One IU neutralizes 10000 LD50 BoNT/A. The botulinum neurotoxin serotype A antitoxin standard was trivalent Botulismus Antitoxin Behring (registration no. 31a/78) Batch 080031A from Novartis Vaccines and Diagnostics GmbH & Co. KG, 35006 Marburg).
We hope that this provides some clarity on the issue. We refrained from explaining the experimental details of the MHDA here because the authors did not perform the assay themselves but had sent it to an independent lab/company.
In order to explain in detail what mIU/ml stands for would require the entire MHDA procedure to be explained in great detail. We do not think that this is necessary here, as our main objective is to document the overall trend and decline in nAbs over time. This can be understood without having to explain how the neutralizing potency of serum dilutions are calculated on the basis of a standardized BoNT/A preparation which is denominated in international (milli) units per ml of serum.
In addition, we have provided additional information on the MHDA, including citations which describe the procedure in great detail (now in lines 810-814) that readers can refer to if more details are required.
We hope that this acceptable for the reviewer.
13) Line 666: the photos are poor quality with insufficient contrast between the background and the patient’s face. Please provide better photos.
Thank you for drawing our attention to this issue. The photos did not turn out as well when inserted into the word document. The original image files in higher quality will be uploaded separately upon acceptance.
14) Line 673, 677 and 683: please use “incobotulinumtoxinA” instead of Xeomin, here and elsewhere.
We thank the reviewer for highlighting this and have replaced all brand names (Xeomin® and Botox®) with their non-branded names (INCO and ONA) throughout the manuscript.
15) Line 677: please stipulate here whether the photos are with maximum frown.
We thank the reviewer for this suggestion. The legend to Fig. 4 (now line 692) has been updated to:
“Photographic demonstration of clinical response to INCO treatment in the glabella at maximum frown: (left) before; September 2023 (right) after; November 2023).”
16) Line 680: please provide some context on why the treating physician continued to treat this patient in the presence of immunoresistance.
We thank the reviewer for the opportunity to clarify on this point. We included more information on why the patient continued to receive treatment despite BoNT/A immunoresistance (now lines 649-653):
“As this patient was unwilling to discontinue BoNT/A treatment completely in the hope that she might become responsive again at some point in time, she continued to receive INCO every 3-4 months, from July 2019 – April 2024, with a dose of 50 units per masseter (total 100 units). This approach allowed the patient’s nAb titers to be monitored closely over time while receiving continous CPF-INCO.”
17) Line 695: please change “does” to “did not, in this case,”
We thank the reviewer for this suggestion and have replaced “does” with “did not in this case” (now in line710).
18) Line 702: please add “presumably” prior to “because”
We thank the reviewer for this suggestion and have added “presumably” prior to “because” (now in line 717).
19) Line 705: Please add “although relatively short observation periods” after “[28]”
We thank the reviewer for this suggestion and have added “notwithstanding the relatively short observation periods of 12 - 16 weeks” to this sentence (now lines 720-721).
We would, however, like to add that the observation periods were 12 to 16 weeks after injections in the TOWER study (Wissel J, et al. Safety and efficacy of incobotulinumtoxinA doses up to 800 U in limb spasticity: The TOWER study. Neurology. 2017 Apr 4;88(14):1321-1328), which would suffice to initiate the production of nAbs after injection of BoNT/A if this were a strong immunogen. Nevertheless, this period of 12 – 16 weeks does not compare to treatments over years and this addition is therefore justified.
We hope that the reviewer is agreeable with our proposed edits.
20) Line 734: The author asks why not all patients respond to INCO. The author needs to acknowledge explicitly that that their hypothesis may not be applicable to most patients. Indeed, clinical experience is that patients who are resistant to one BoNTA are resistant to all BoNTAs, including INCO.
We thank the reviewer for this comment and the opportunity to clarify this point.
First, we agree with the reviewer’s comment that patients who have developed complete immunoresistance to one BoNT/A preparation will also be resistant to other BoNT/A preparations as nAbs bind to the core neurotoxin molecule, which is similar in all preparations. This is why patients with complete SNR secondary to nAb formation will not immediately respond when they are switched to INCO until nAb titers decline to a clinically irrelevant levels, as illustrated in our case example. Indeed, our proposed approach to continue treatment with a CPF-BoNT/A preparation e.g., INCO is only likely to work in cases with partial SNR.
A key objective of this manuscript is to provide a scientific explanation for the results that Hefter et al. had reported in their cohort of patients with partial SNR due to nAb formation. Hefter et al. reported that nAb titers remained unchanged in 16% of cases, dropped below the initial titers at the end of the observation period in 84% of cases, and even reached the lower detection limit of the MHDA or became negative in 62% of cases. The results from this study support that switching to INCO is a viable option in the majority of patients with partial SNR, but there remains a smaller proportion in whom this approach might not be applicable and this is difficult to predict. We have made minor edits to make it clearer that this hypothesis is likely to be applicable only to partial SNR cases (now lines 745-750):
“This novel approach to manage nAb-mediated SNR with a low immunogenic BoNT/A product seems to be quite attractive for some patients, but it has limitations. One limitation of this approach is that it is impossible to predict which patient with partial immunoresistance will respond. Continuation of treatment with low immunogenic BoNT/A does not work in all patients, as evident in both studies by Hefter, et al.”
Complete SNR cases do not immediately respond to CPF-INCO in the presence of a clinically relevant titer of anti BoNT/A nAbs, but as we try to explain in this manuscript, preventing the reactivation of memory B cells with a CPF-BoNT/A product may result in a gradual decline of nAb titer, eventually reaching a time point when INCO works again without boostering the immune system. However, it is difficult to predict how long it will take to reach this state as discussed in the subsequent paragraph of this manuscript (now lines 778 – 789). Continuing treatment with a CPC-BoNT/A product would have resulted in boostering and continuous restimulation of immunological memory.
Furthermore, we have added “some” in line 844 that now reads:
“This explains why continuation of treatment with CPF-INCO may be a viable treatment option for some patients in the presence of nAbs.”
We hope that the reviewer finds our explanation and amendments acceptable.
21) Line 776: please reflect that one of the potential issues is related to increased immunogenic response. Consequently, “…this translates to potentially increased stimulation of the immune system, unnecessary…” This is supported by the author’s statement later in the paragraph with the statement “…most likely due to a boostering of memory B cells despite the lack of immunogenic adjuvants.”
We thank the reviewer for emphasizing this important point once more. This issue is addressed further down in the text (now lines 799 – 802):
“In such a situation, continuation of treatment with CPF-INCO might prolong the period of non-responsiveness rather than shorten it as compared to immediate cessation of treatment. nAb titers did not drop in 6 out of 37 patients over the study period of more than four years [36].”
In addition, we have also amended the text to reflect this point as suggested (now lines 790-793):
“Given these limitations, it seems unrealistic and impractical to prophylactically continue BoNT/A treatment with a low-immunogenic product in all patients with nAb-mediated SNR, especially those with complete SNR, as this translates to unnecessary injections, high costs, and potentially prolonging the period of non-responsiveness.”
We hope that the reviewer finds this acceptable.
22) Line 794: The author states “The ex vivo MHDA… is very reliable”. Please cite an assessment of the reliability of the MHDA. Are the false positives and negatives with the MHDA known when comparing to clinical outcomes?
We thank the reviewer for highlighting this and have added the following text (now in lines 810-814):
“The ex vivo MHDA is extremely sensitive in detecting nAbs to BoNT/A [reviewed in 4, 136,137]. This assay has demonstrated a very high sensitivity compared to other detection methods e.g., mouse protection assay [reviewed in 4,38,138], but also carries the risk of a higher false-positive rate due to the possibility that low nAb titers of unclear clinical relevance may be detected.”
As it was not our intention to compare different assay systems in this manuscript, we restricted ourselves to this short statement in the hope that the reviewer will accept it. In general, the authors have the impression that the MHDA is presently accepted widely as being the most useful ex vivo assay to measure neutralizing antibodies for BoNTA with the least animal sacrifices.
23) Line 797: “detect Abs” – are they referring to neutralizing or binding antibodies?
We thank the reviewer for highlighting this and apologize for the oversight.
For clarity, we have revised the text to (now in lines 816-817):
“An ideal test would be a stick assay that allows physicians to conveniently detect anti BoNT/A Abs, both non-neutralizing and neutralizing Abs, in their clinical practice.”
We would also like to draw the reviewer’s attention to the fact that we had explained this point a little bit further down in the text (now lines 821-823):
“A major limitation of a stick assay and/or an ELISA, in contrast to the MHDA, is their inability to discriminate between non-neutralizing and neutralizing anti-BoNT/A Abs.”
24) Line 805: since an ELISA binding antibody assay is not diagnostic, but used as a screening assay, the MHDA is the key confirmatory step. The author should stipulate that the MHDA, or a comparable bioassay (e.g., the mouse neutralization assay), is the “critical” or “key” next step rather than “can be the next step. In the discussion of the MHDA, the author should further stipulate and cite the several studies that have reported that a positive nAB does not necessarily correlate to, nor predict SNR.
We thank the reviewer for this suggestion. Although an ELISA can in principle be used for diagnostic purposes, we do agree that the MHDA is critical for the diagnosis of nAb-mediated SNR. In combination with clinical observations of signs and symptoms of SNR, it is possible for physicians to confirm the diagnosis of nAb-mediated secondary non-response or immunoresistance. Therefore, we rephrased the text (now lines 824-829):
“Upon the identification of BoNT/A Abs, MHDA serve as a critical second step to distinguish nAbs from non-neutralizing Abs. It should be noted here that several studies have reported that a positive detection of nAbs does not necessarily correlate or predict SNR [41,142-144]. However, the positive detection of nAbs on MHDA correlated clinically with signs and symptoms of SNR will allow physicians to confirm the diagnosis of nAb-mediated SNR.”
25) Line 823ff: In the author’s summary, the author should acknowledge the other limitations of the practical application of their approach. This includes the potential to further boost the immune response.
We thank the reviewer for raising this important point, especially in the summary.
We have amended the text which now reads (now lines 847-848):
“Presently, however, the lack of predictability of when is the best time-point to inject a low immunogenic BoNT/A product into which patient without further boostering the BoNT/A-specific immune response limits the practical application of this approach.”

Round 3
Reviewer 1 Report
Comments and Suggestions for Authors
Thanks to the authors for addressing the comments. I appreciate their taking the time to improve the paper. There are just 3 additional comments that should complete the review cycle.
- Line 470: Please change "will not be taken up" to "should not be taken up"
- Line 827: Please add citation 110
- Line 828: "...the positive detection of nAbs on MHDA correlated clinically...". Pleaes update with "...the positive detection of nAbs on MHDA, when correlated clinically with signs and symptoms of SNR, will allow..."
